# Relativistic magnetic reconnection driven by a laser interacting with a micro-scale plasma slab

Longqing Yi [1,2], Baifei Shen [3], Alexander Pukhov [4] & Tünde Fülöp [1]

Magnetic reconnection (MR) is a fundamental plasma process associated with conversion of the magnetic field energy into kinetic plasma energy, which is invoked to explain many non-thermal signatures in astrophysical events. Here we demonstrate that ultrafast relativistic MR in a magnetically dominated regime can be triggered by a readily available (TW-mJ-class) laser interacting with a micro-scale plasma slab. Three-dimensional (3D) particle-in-cell (PIC) simulations show that when the electrons beams excited on both sides of the slab approach the end of the plasma, MR occurs and it gives rise to efficient energy dissipation that leads to the emission of relativistic electron jets with cut-off energy ~12 MeV. The proposed scenario allows for accessing an unprecedented regime of MR in the laboratory, and may lead to experimental studies that can provide insight into open questions such as reconnection rate and particle acceleration in relativistic MR.

[1] Department of Physics, Chalmers University of Technology, 41296 Gothenburg, Sweden. [2] State Key Laboratory of High Field Laser Physics, Shanghai Institute of Optics and Fine Mechanics, Chinese Academy of Sciences, P.O. Box 800-211201800 Shanghai, China. [3] Department of Physics, Shanghai Normal University, 200234 Shanghai, China. [4] Institut für Theoretische Physik I, Heinrich-Heine-Universität Düsseldorf, 40225 Düsseldorf, Germany. Correspondence and requests for materials should be addressed to L.Y. (email: longqing@chalmers.se)

Many high-energy astrophysical environments are strongly magnetized, that is, the magnetic energy per particle exceeds the rest mass energy, so that the magnetization parameter $\sigma \equiv B_0^2/4\pi nm_ec^2 \geq 1$, where $B_0$ is the magnetic field strength, $m_e$ is the electron mass, $n$ is the plasma density, and $c$ is the vacuum light velocity. In such environments, magnetic reconnection (MR) operating in the relativistic regime, plays a key role in the transfer of large amounts of magnetic to kinetic energy via field dissipation[1–3]. Motivated by numerous astrophysical observations, such as high energy emission in pulsars[4], cosmological gamma-ray bursts[5], and active galactic nuclei jets[6,7], the study of relativistic MR has made rapid progress in the last few decades through analytical studies[8,9], as well as 2D[10–13] and 3D[14–18] PIC (particle-in-cell) simulations. However, due to difficulties in achieving the extreme magnetic energy densities that are required to observe relativistic MR in laboratory environments, previous experimental studies investigated mainly the non-relativistic regime ($\sigma < 1$). These include experimental observation of MR in tokamaks[19] or dedicated experiments, such as MRX (Magnetic Reconnection Experiment)[20].

High-intensity laser–plasma interaction[21–27] is a promising way to break through the relativistic limit as the energy densities that can be achieved by high-intensity laser facilities worldwide[28,29] are rising rapidly. These facilities provide an important platform for the study of relativistic MR and therefore attract extensive attention. In the typical laser-driven MR experiments, two neighboring plasma bubbles are created by laser–matter interaction. The opposite azimuthal magnetic fields arise due to the Biermann battery effect ($\nabla n \times \nabla T_e$, where $T_e$ is the electron temperature)[30] and reconnect in the midplane as they are driven together by the frozen-in-flow of bulk plasma expansion[21–27,31,32].

Most of the previous laser-driven MR studies are focused on the non-relativistic case, and the ratio of the plasma thermal and magnetic energy density, $\beta$, is typically high ($\beta > 1$)[33]. Recently, it has been reported that relativistic MR conditions could also be achieved with such a scenario[34], where subpicosecond high-intensity $10^{19}$ W cm$^{-2}$ laser pulses are used to create ~100 MG magnetic fields. However, at such intensities, the heating of hot electrons by laser pulse becomes very efficient[35,36]. An empirical scaling law $\eta \approx 1.2 \times 10^{-15} I^{0.74}$[37] indicates the ratio of laser energy converted to the hot electrons ($\eta$) approaches up to 50% as the laser intensity reaches $5 \times 10^{19}$ W cm$^{-2}$, where $I$ is the laser intensity in unit of watts per square centimeter.

Due to the efficient laser heating, achieving relativistic reconnection in a magnetically dominated plasma ($\beta < 1$) is very challenging, and it is therefore not clear what role MR plays in terms of energy balance. Recently it has been reported that magnetically dominated MR can be achieved by a double-turn Helmholtz capacitor-coil target[38], but this approach is very difficult to extend to the relativistic regime because it requires a kJ-class laser system. Thus, in spite of the remarkable progress that has been made, relativistic MR in low-$\beta$ environments ($\beta < 1$), which is closely related to the interpretation of many space plasma measurements and astronomical observations[39], has not been thoroughly studied.

In this paper, we propose a experimental setup based on the interaction of a readily available moderately intense (TW-mJ-class) laser with a micro-sized plasma slab. The fast electrons driven by the laser pulse produce 100-MG-class magnetic fields, and the magnetic reconnection takes place when they are driven together by the attraction between the associated strong currents. In the meantime, the laser-heating effect in the reconnection layer is dramatically reduced by using side incidence, thus allowing the access of the magnetically dominated regime ($\beta < 1$) for relativistic reconnection ($\sigma > 1$). This potentially opens a realm to the

experimental study of an interesting regime of MR. Using 3D PIC simulations, a comprehensive numerical experiment is presented, which demonstrates the MR event observed in the proposed scheme has a significant effect on the whole system. It leads to fierce (0.1-TW-class) field dissipation and highly efficient particle acceleration, which covers 20% of the total energy transition.

The proposed setup can provide an important experimental platform for studying the energetics in relativistic MR, in particular particle acceleration processes and the resulting power-law electron spectrum. Most of the previous studies in this regime are based on simulations which use highly idealized initial conditions, such as Harris-type current sheet[40] and periodic boundary conditions. Recent simulations without preformed current sheets result in significantly different electron spectra[41]. This could be hard to probe in a high plasma $\beta$ environment as the particle energization by MR can then be very small compared to other mechanisms (laser heating for example). In addition, other aspects of relativistic MR, such as kinetic beaming[18] and physics related to strongly driven reconnection[12] could potentially be studied with the proposed scheme.

In this paper, we mainly focus on the description of our setup and the associated MR signatures including intense relativistic jets, quantified agyrotropy peaks in the diffusion area[42], and out-of-plane quadrupole field structures[43]. A discussion on the magnetic energy dissipation and particle acceleration is also presented. With recent advances in laser pulse cleaning techniques[44–46] and micro-target manufacturing[47], the proposed scenario is very promising to be implemented in experiments.

## Results

**Generation of relativistic jets.** A sketch of the simulation setup is shown in Fig. 1a. A linearly polarized (in $y$ direction) laser with normalized laser intensity $a_0 = eE_0/m_ec\omega_0 = 5$ (intensity ~$3 \times 10^{19}$ W cm$^{-2}$) propagates along the $x$-axis, where $E_0$ is the laser amplitude, $\omega_0 = 2\pi c/\lambda_0$ and $\lambda_0 = 1$ µm are the frequency and wavelength of the laser, respectively. The laser spot size is $4\lambda_0$ and the duration is $15T_0$, where $T_0 \approx 3.3$ fs is the laser cycle. A plasma slab with thickness (in the laser-polarizing direction) $d = 1\lambda_0$ and length (in the laser propagation direction) $L = 20\lambda_0$ splits the laser pulse in half. The main part of the slab has an uniform density of $20n_c$, where $n_c = m_e\omega_0^2/4\pi e^2 = 1.1 \times 10^{21}$ cm$^{-3}$ is the critical density. At the end of the structure (coronal region, represented by the area within the blue-box framework in Fig. 1), the density drops exponentially as $x$ increases (scale length $l = 2\lambda_0$). As the laser pulse sweeps along the slab, it drives two energetic electron beams on both sides of the plasma surface[48]. These electron beams are typically overdense[49] and capable of generating 100 MG level opposing azimuthal magnetic fields in the middle, as shown by the black arrows in Fig. 1a. Detailed parameters of the simulation can be found in Methods. Note that, in the 3D PIC simulations presented below we use a reduced $20n_c$ density for computational efficiency, as well as a symmetric configuration and sharp plasma-vacuum boundary. However, as discussed in Supplementary Notes 2 and 3, the underlying physical process, the interaction of two laser-driven electron beams triggering MR, is a very robust mechanism that does not depend on these conditions.

In the early stage (before the electron beams reach the corona), MR does not occur because a strong return current is excited inside the slab, which separates the antiparallel magnetic fields on each side of the slab, and the magnetic energy remains constant (after the initial rise due to the laser–matter interaction) during this period. As the electron beam approaches the end of the

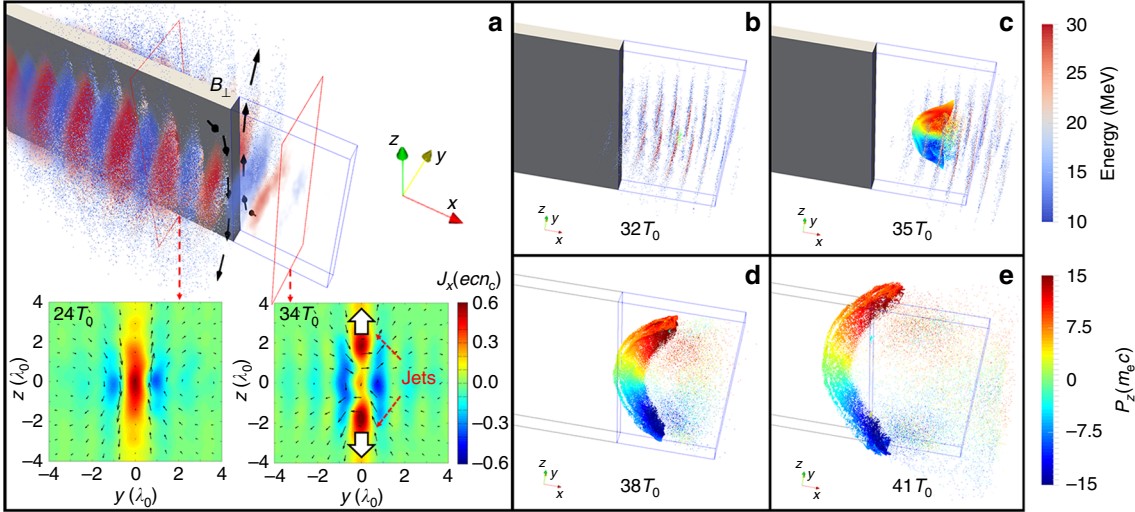

**Fig. 1** Schematic of the proposed setup and relativistic jets generation. **a** A moderately high-intensity laser pulse ($a_0 = 5$) propagates along the $x$-direction, and is splitted in half by a micro-sized plasma slab. The laser drives two energetic electron beams on both sides of the plasma surfaces, which generate 100 MG level opposing azimuthal magnetic fields in the middle. Ultrafast magnetic reconnection is observed as the electron beams approach the coronal region (the area within the blue box, where the plasma density decreases exponentially) at the end of the slab. The two insets below show the transverse magnetic fields (black arrows) and longitudinal electric current density (color) at the cross-section marked by the red rectangle (separated by $10\lambda_0$) at simulation times $t = 24T_0$ and $t = 34T_0$, respectively. **b–e** Generation and evolution of the relativistic jet resulting from MR at times $32T_0$, $35T_0$, $38T_0$, and $41T_0$, respectively. The rainbow color bar shows the transverse momentum $P_z$ of the jets formed by the background plasma electrons in **b–e**, and the blue-red color bar shows the energy of the electron bunch driven by the laser pulse in **b**, **c**

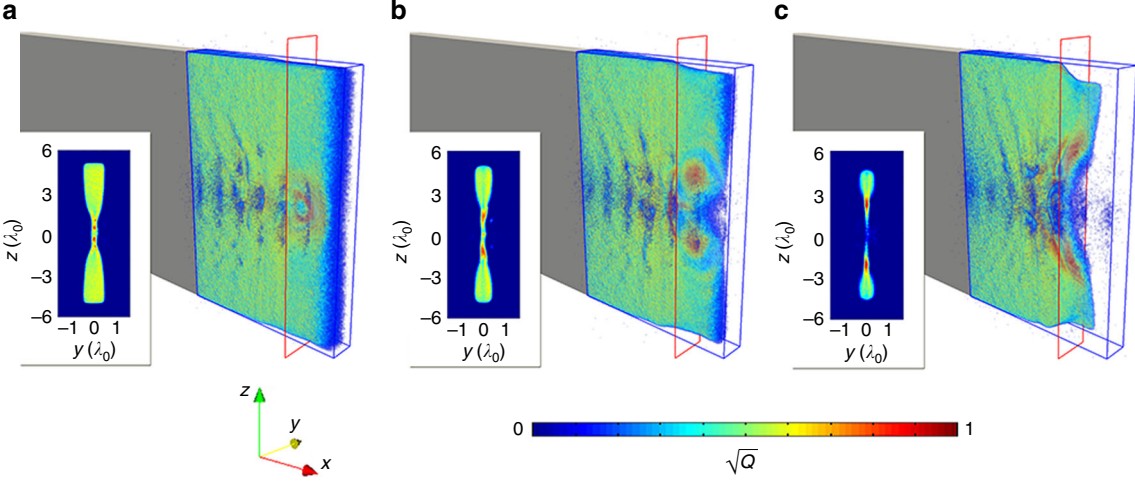

**Fig. 2** Gyrotropy quantification at different times. Square root of quantified pressure tensor agyrotropy $\sqrt{Q}$ in the coronal plasma at simulation time $t = 32T_0$(**a**), $33T_0$(**b**), and $34T_0$(**c**). The insets show the value of $\sqrt{Q}$ at the cross-section with longitudinal coordinate $x = 26\lambda_0$, which is marked by the red rectangles in **a–c**

structure, the plasma density decreases rapidly so that the electron number in the local plasma becomes insufficient to form a return current that is strong enough to separate the magnetic fields. Therefore, due to Ampère's force law, the electron beams on both sides attract each other and flow into the mid-plane coronal plasma. The magnetic field lines that move with the electron beams are pushed together and reconnect. The reconnection magnetic field in the corona is ~100 MG. An X-point magnetic field topology is observed as shown by the right-bottom inset of Fig. 1a.

As the field topology changes, the explosive release of magnetic energy results in the emission of relativistic jets as shown by Fig. 1b–e. These jets are formed by the background plasma electrons in the corona, which start to appear at approximately $t$ = $32T_0$ [Fig. 1b] and acquire relativistic energies within a few laser cycles. They propagate backwards ($-x$ direction) toward the exhaust region of the reconnection site ($\pm z$ direction). The backward longitudinal momentum stems from efficient acceleration due to magnetic energy dissipation as will be discussed in the remainder of this work. The electrons in the jets distinguish themselves from the rest of background electrons that are heated by the laser pulse in this region by remarkably higher energies (top 0.2% on the electron spectra, with mean energy $\overline{E} \sim 4.7$ MeV), and considerably small $y$-divergence ($\theta_y = |P_y|/P \sim 0.1$, where $P$ is the electron momentum and $P_y$ denotes its $y$-component). These electrons form dense jets, with the density $n_{jet} \sim 5n_c$ initially, however, it decreases rapidly due to dispersion in the $z$-direction. The total charge of the jets is ~0.2 nC.

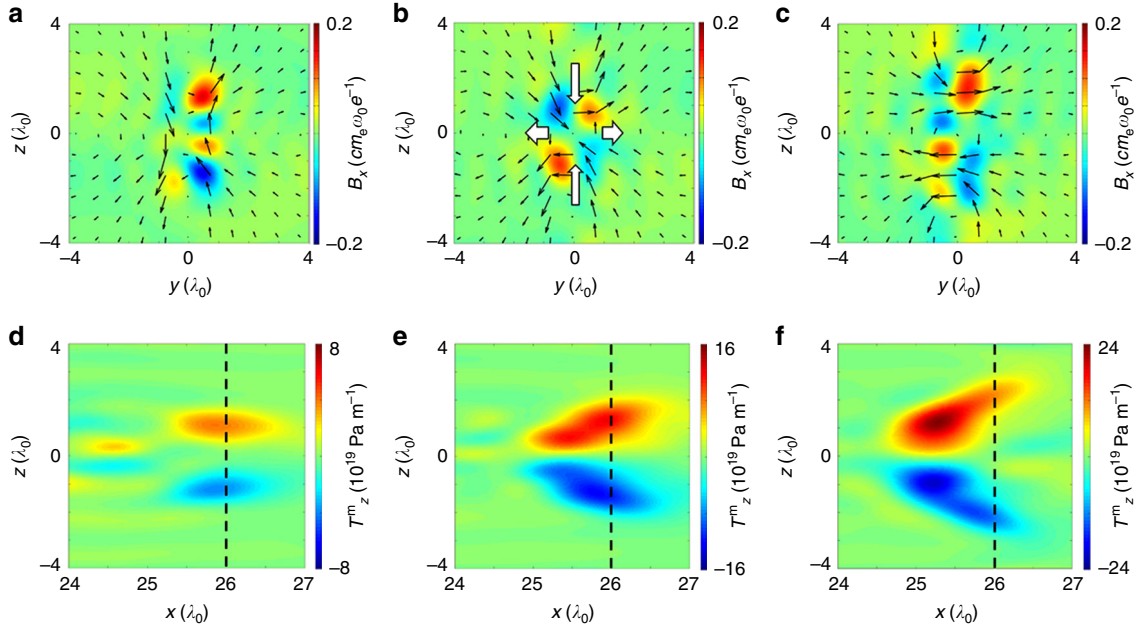

**Fig. 3** Evolution of magnetic fields and magnetic tension force during the reconnection. **a–c** Static magnetic fields (frequency below $0.8\omega_0$) and **d–f** z-component of magnetic tension force at simulation time $t = 32T_0$ (**a, d**), $33T_0$ (**b, e**), and $34T_0$ (**c, f**). In **a–c** the transverse ($B_y$, $B_z$) and longitudinal ($B_x$) components of magnetic field are presented by the black arrows and color, respectively. The bold white arrows in **b** show the inflow (horizontal) and outflow (vertical) electric currents that result from Hall reconnection. The black-dashed lines in **d–f** mark the cross-section where the corresponding magnetic fields (**a–c**) are shown

**Pressure tensor agyrotropy and observation of field-line rearrangement**. In order to locate the reconnection site in the corona, we calculate the scalar measure of the electron pressure tensor gyrotropy that was suggested recently by Swisdak[42],

$$Q = \frac{\mathcal{P}_{12}^2 + \mathcal{P}_{13}^2 + \mathcal{P}_{23}^2}{\mathcal{P}_\perp^2 + 2\mathcal{P}_\perp \mathcal{P}_\parallel}. \quad (1)$$

Here $\begin{pmatrix} \mathcal{P}_\parallel & \mathcal{P}_{12} & \mathcal{P}_{13} \\ \mathcal{P}_{12} & \mathcal{P}_\perp & \mathcal{P}_{23} \\ \mathcal{P}_{13} & \mathcal{P}_{23} & \mathcal{P}_\perp \end{pmatrix}$ is the electron pressure tensor

$$\mathbb{P} = m_e \int (\mathbf{v} - \overline{\mathbf{v}})(\mathbf{v} - \overline{\mathbf{v}}) f d^3\mathbf{v} \quad (2)$$

transformed into a frame in which the diagonal components are in gyrotropic form, that is, one of the coordinate axes points in the direction of the local magnetic field and the others are oriented such that the final two components of the diagonal of $\mathbb{P}$ are equal (see the Appendix in ref. [42] for details). $f(x, y, z, \mathbf{v})$ is the distribution function at position $(x, y, z)$ and velocity $(\mathbf{v})$, and $\overline{\mathbf{v}}$ is the mean velocity. It should be noted that although Eq. 2 is the non-relativistic definition of the electron pressure tensor, the relativistic effects are not expected to have a significant influence here since more than 99% of the electrons in the corona are non-relativistic, namely $\gamma - 1 < 1$, where $\gamma = 1/\sqrt{1 - (v/c)^2}$ is the Lorentz factor of the electrons. In general, for gyrotropic electron pressure tensors $Q = 0$, and the maximum departure from gyrotropy is $Q = 1$. High values of $Q$ usually identify regions of interesting magnetic topology, such as separatrices and X-points in magnetic reconnection.

Figure 2 shows that the space and time where $Q$ reaches its peak in the 3D PIC simulation coincides with the appearance of the electron jets shown in Fig. 1b, c. The slab is significantly pinched during the interaction[50], which results in a higher inflow velocity and thus a faster reconnection rate. Knowing the location of the reconnection site allows one to calculate the magnetization parameter. By substituting the electron density and magnetic field

for the region with peak agyrotropy, one obtains the maximum magnetization parameter $\sigma_{max} \approx 30$. Therefore we conclude that the observed MR is in the relativistic regime. In addition, by calculating the time dependence of the magnetic flux ($\psi$) at the reconnection site at $z = 0$, one obtains the average reconnection rate from $t = 32T_0$ to $t = 34T_0$ to be $\overline{d\psi/dt} = 0.32cB_0$, with a maximum value $0.47cB_0$ at $t = 33.4T_0$.

It is generally understood that the jets are accelerated out of the reconnection region by the magnetic tension forces ($\mathbf{T}^m = (\mathbf{B}\cdot\nabla)\mathbf{B}/4\pi$) of the newly connected, strongly bent magnetic field lines. In Fig. 3, we plot the magnetic fields at $x = 26\lambda_0$, and the z-component of the magnetic tension force $T_z^m$ at the midplane ($y = 0$). A quadrupole longitudinal magnetic field pattern emerges as the MR occurs, which is indicative of Hall-like reconnection[43], where the reconnection rate is significantly enhanced due to decoupling of electron and ion motion. The amplitude of $B_x$ is 20 MG, which is ~20% of the reconnected magnetic field.

Figure 3 illustrates that an enormous magnetic tension force is generated in the corona because of the reconnection. The pressure exceeds the relativistic light pressure ($P = I/c$ with $a_0 \sim 1$) within one laser wavelength, which exerts a strong compression of the electron jets that results in the observed high-density emission. Moreover, the shape of magnetic tension in Fig. 3 shows the dynamics of the newly connected field lines strengthening and relaxing, during which the energy is transferred to the plasma particles. The phenomenon is consistent with the shape and emission direction of the jets that we observed in Fig. 1c–e.

**Magnetic energy dissipation and particle acceleration**. In order to gain a deeper understanding of energy transfer in the relativistic MR, we now focus on the field dissipation process. The observed dissipation power is of the order of 0.1-TW, which results in a highly efficient energy transfer from the magnetic fields to the kinetic energy of plasma. Figure 4a is a graphic demonstration of 3D field dissipation, where the work done by the longitudinal electric field per unit volume and unit time ($E_x J_x$)

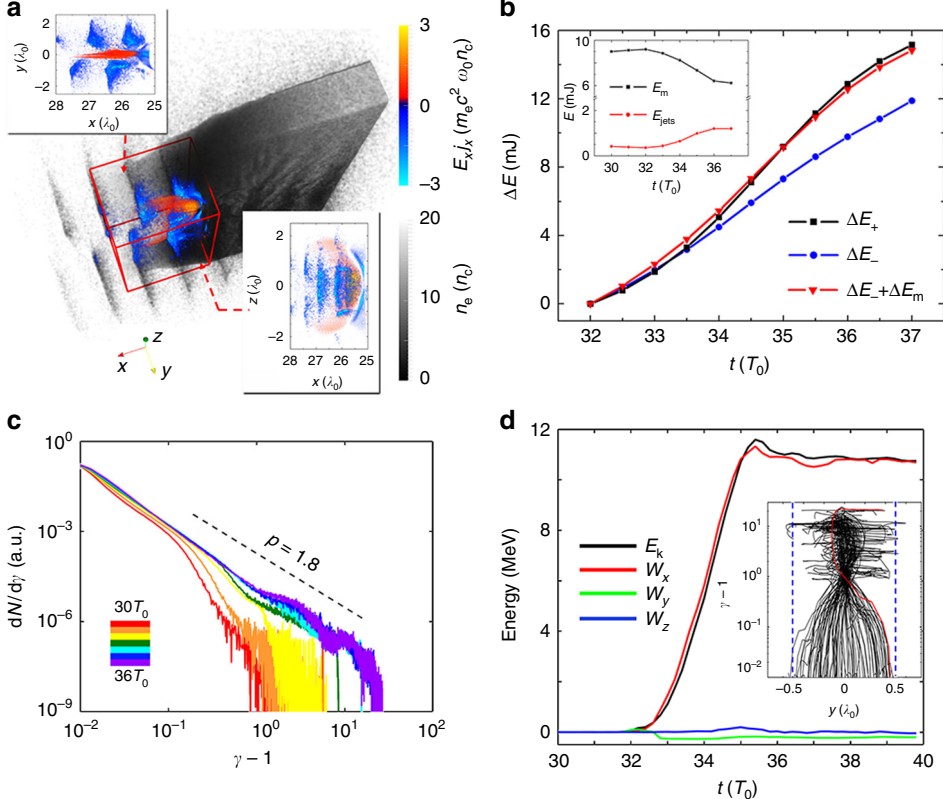

**Fig. 4** Magnetic energy dissipation and the energization of non-thermal electrons. **a** Field dissipation ($E_x j_x$) and electron density at $t = 33T_0$ in the corona, the insets represent the top and side views of $E_x j_x$ in the reconnection site (marked by the red box). **b** Time dependence of total energy increase in electrostatic fields, electrons in the corona, and protons ($\Delta E_+$), energy reduction of electromagnetic fields and other electrons ($\Delta E_-$), as well as the total energy reduction that includes magnetic field dissipation ($\Delta E_- + \Delta E_m$), inset shows the evolution of static magnetic energy $E_m$ and total kinetic energy of electron jets. **c** Coronal electron spectra from $30T_0$ to $36T_0$. **d** The temporal evolution of the kinetic energy ($E_k$) and the work done by each electric field component ($W_x$, $W_y$, and $W_z$) for one representative electron. The inset plane shows the phase-space trajectory ($\gamma - 1$ plotted vs. $y$) of the total 100 tracked electrons, where the blue-dashed line marks the boundary of plasma slab and the trajectory in red represents the case shown in **d**

is presented. The blue chains on both sides of the slab show the process of the laser-driven electron-beams losing energy to the static electric field, which is sometimes referred to as enhanced target normal sheath acceleration[51]. Meanwhile in the midplane, one can see that the energy flows are directed in the opposite direction as MR occurs, the electrons in the coronal plasma collectively extract energy from the (reconnection associated) electric fields and thus gain kinetic energy. To further clarify the dissipation process, we also conducted a comparison simulation which is presented in the Supplementary Note 1.

To further understand the role that relativistic MR plays in the energy transfer process, we calculated the energy change for each component in the simulation, that is, laser pulse (frequency ≥ $0.8\omega_0$), static electric/magnetic field (frequency < $0.8\omega_0$), and kinetic energy of electrons and protons), and the results are shown in Fig. 4b. Let $\Delta E_+$ denote the total energy increase in the electrostatic field, coronal electrons and protons, $\Delta E_-$ denote the energy reduction in the laser pulse and other electrons (mostly laser-driven electron beams), and $\Delta E_m$ represent the energy loss in the static magnetic fields. As one can see, a significant contribution to the total energy transfer comes from the annihilation of magnetic fields due to relativistic MR, which accounts for ~20% of the total energy transition. Also, Fig. 4b indicates the static magnetic field loses ~3.0 mJ energy in five laser cycles, which yields an efficient magnetic energy dissipation with power ~0.18 TW. In addition, the evolution of the static magnetic energy and the kinetic energy of the electron jets are plotted in the inset of Fig. 4b. A good time synchronization is

observed between the magnetic field dissipation and electron acceleration. It should be noted the target normal electrostatic fields also play a role in the electron acceleration. However, judging from the peak reconnection rate $E_r = 0.47cB_0 \approx 1.5$ TV m$^{-1}$ and peak electric field $E_{max} \approx 5.8$ TV m$^{-1}$, a significant (26%) energy gain is attributed to the reconnection fields, which will provide a clear experimental signature in the electron spectrum.

In Fig. 4c, we plot the energy spectra of the electrons initially located in the corona, from $30T_0$ to $36T_0$. One can see that as the reconnection occurs, the released magnetic energy is transferred to the non-thermal electrons. The total electron kinetic energy in the corona has increased by a factor of 4 during the reconnection and a hard power-law electron energy distribution $dN/d\gamma \propto 1/\gamma^p$ is obtained with index $p \approx 1.8$. The low energy part of the electron spectra cannot be fitted by a simple thermal distribution because the laser–plasma interaction gives rise to a two-temperature energy distribution[52]. In order to estimate the plasma $\beta$ in the reconnection layer, we calculate the average electron kinetic energy in the corona at $t = 34T_0$, $\overline{E}_k \sim 38$ keV. Using the reconnection magnetic field ~100 MG and electron density ~$1n_c$, the plasma $\beta$ is ~0.15. As the electrons are gaining energy during the reconnection process, the value of $\beta$ increases with time, but it does not change significantly ($\beta \approx 0.07$–$0.22$ for $t = 32$–$36T_0$), and during the whole reconnection process the adjacent plasma is maintained in the magnetically dominated regime.

To reveal the mechanism of particle acceleration, we randomly track 100 electrons that attain relativistic energy during the MR, and display one representative case in Fig. 4d, where the electron

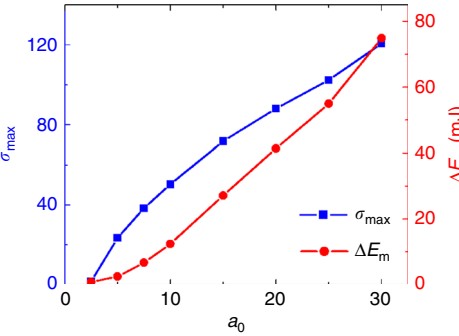

**Fig. 5** Relativistic reconnection with various laser intensities. A parameter scan showing the dependence of the maximum magnetized parameter $\sigma_{max}$, and the magnetic energy dissipation $\Delta E_m$, on the normalized laser amplitude $a_0$

kinetic energy and work done by each component of the electric fields are plotted as a function of time. It illustrates that the electric field associated with reconnection at the X-point is primarily responsible for the energization of the electrons. This is supported by the trajectories of the tracked electrons in phase space (shown by the inset of Fig. 4d, where the trajectory of the representative electron for the energy plotting is drawn in red), which shows that almost all the electrons gain their energies within a narrow plane ($-0.1\lambda_0 < y < 0.1\lambda_0$) adjacent to the X-point. In this region, since the magnetic field vanishes due to the reconnection, the electrons become unmagnetized and can be accelerated freely. Moreover, as indicated by the horizontal lines in the phase space, once the electrons escape from this narrow plane, the acceleration process stops immediately. Recent studies[31,32] have pointed out that the randomness of the electron injection and escape from the acceleration region give rise to the observed power-law energy distribution as shown in Fig. 4c.

## Discussion

The present work demonstrates that the interaction of a high-intensity laser and a micro-scale plasma can trigger relativistic MR, which leads to highly efficient magnetic energy dissipation and gives rise to intense relativistic jets. In order to explore the potential of the proposed scenario, we conducted a series of 3D PIC simulations to study the laser intensity dependence of the maximum magnetization parameter $\sigma_{max}$ and the magnetic energy dissipation $\Delta E_m$. The results, displayed in Fig. 5 show that, by applying a micro-sized slab target, relativistic MR can be accomplished at approximately $a_0 = 2.5$ (where $\sigma_{max} = 1.7$), with a laser energy of only 50 mJ. Such laser systems are readily available worldwide, which may open a way to extensive experimental study of relativistic MR and greatly advance our knowledge of the fundamental processes, such as particle acceleration and magnetic energy dissipation. Nevertheless, it is worth noting that the system size in the proposed setup is mainly determined by the target thickness and laser spot size. A small system size can help to achieve a high energy density but is not mandatory. For the main simulation presented in this paper, the system size is ~4 μm, which is about $8d_e$ and $25\rho_0$, where $d_e$ and $\rho_0$ are the electron inertial length and electron gyro-radius, respectively. A larger system size may lead to other interesting phenomena[53] as well as different particle energization[54]. However, this is beyond the scope of the current paper.

Figure 5 illustrates that as the incident laser intensity grows, the rate at which $\sigma_{max}$ increases gradually slows down after the initial fast-growing phase ($3 < a_0 < 10$), while an opposite evolution is observed for $\Delta E_m$. This is because the reconnection site moves toward the high-density region as the laser-driven electron beams

become increasingly intense, which results in stronger field dissipation (more electrons gain energy from the reconnecting field). On the other hand, the enhancement of the local plasma frequency $\omega_p$ slows down the growth of $\sigma_{max}$. Nevertheless, it is shown that for the normalized laser amplitude $a_0 = 30$ ($I \sim 10^{21}$ W cm$^{-2}$), the regime of $\sigma_{max} \sim 120$ can be accessed and the released magnetic energy due to field dissipation is as high as 75 mJ. With next-generation Petawatt laser facilities, such as ELI[28], these results might open a realm of possibilities for experimental studies of laboratory astrophysics and highly efficient particle acceleration induced by relativistic reconnection.

## Methods

**PIC simulation.** The 3D PIC simulations presented in this work were conducted with the code EPOCH[55]. For the primary simulation we presented in this work, the dimensions of the simulation box are $x \times y \times z = 30\lambda_0 \times 15\lambda_0 \times 15\lambda_0$, which is sampled by $1200 \times 600 \times 600$ cells with five macro particles for electron species and three for proton species in each cell. The output of this high-resolution simulation was used to produce Figs. 1, 2, 3 and 4a. The rest of the results presented in this work are conducted with a larger simulation box $x \times y \times z = 40\lambda_0 \times 20\lambda_0 \times 20\lambda_0$ to ensure the laser pulse and electron beams do not leave the simulation box while we analyze the magnetic field energy, but sampled with a lower resolution $800 \times 400 \times 400$ to reduce the computational time. The initial plasma Debye length in the primary simulation is $\lambda_D = 1.6 \times 10^{-7}$ cm. A high-order particle shape function (fifth order particle weighting) is applied to reduce numerical self-heating instabilities (see Sec. 5.1 of ref. [55] for details). The numerical convergence has been confirmed by comparing the physical quantities of interest for the simulations with different resolutions.

In this study we restrict the simulations to the collisionless case, which is justified by the high temperature (~10 keV) achieved in laser–plasma interactions, leading to particles having mean free paths larger than the system size.

**Laser-plasma parameters.** A moderately high-intensity laser beam with normalized amplitude $a_0 = 5$, and a focus spot of $4\lambda_0$ is used to drive the relativistic MR. The temporal profile of the laser pulse is $T(t) = \sin^2(\pi t/\tau)$, where $0 \leq t \leq \tau = 15T_0$, The power and energy of laser pulse are approximately 12 TW and 200 mJ, respectively. The plasma slab dimension is $x \times y \times z = 20(7)\lambda_0 \times 1\lambda_0 \times 10\lambda_0$, where the number in the bracket in x direction denotes the length of the coronal region. The slab is pre-ionized (proton-electron plasma, and physical mass ratio $m_p/m_e = 1836$ is used), the initial temperature is $T_e = T_p = 1$ keV. The plasma density is uniform ($n = n_0$) for the main part of the slab, and decreases exponentially with x in the coronal region, $n = n_0 \exp(-(x - x_0)^2/2l^2)$, where $x \geq x_0 = 21\lambda_0$ and $l = 2\lambda_0$ is the scale length. In most of the simulations we present in this work (except for Fig. 5), $n_0 = 20n_c$ is applied, while in Fig. 5, we use $n_0 = 50n_c$ for the scan runs in order to avoid problems caused by self-induced transparency at ultrahigh laser intensities. The ultra-short laser pulse duration and the relatively low plasma density are used to improve computational efficiency, which may cause slight differences in the simulation outputs, but do not crucially alter the underlying physics of relativistic MR.

**Data availability.** The data that support the findings of this study are available from the corresponding author upon request.

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

## Acknowledgements

The authors would like to thank J. Tenbarge, I. Pusztai, S. Newton, T. Dubois, and the rest of the PLIONA team for fruitful discussions. This work is supported by the Knut and Alice Wallenberg Foundation, the European Research Council (ERC-2014-CoG grant 64712), National Natural Science Foundation of China (No.11505262, No.11335013), and the Strategic Priority Research Program of Chinese Academy of Sciences (Grant No. XDB16). The simulations were performed on resources at Chalmers Centre for Computational Science and Engineering (C3SE) provided by the Swedish National Infrastructure for Computing.

## Author contributions

L.Q.Y. designed and conducted the simulations and analyzed the results, under supervision of T.F. L.Q.Y. and T.F. wrote the paper with contributions from B.F.S. and A.P.

## Additional information

**Competing interests:** The authors declare no competing interests.

