## [Peer Review File · Nature Communications]

Reviewers' comments:

Reviewer #1 (Remarks to the Author):

The authors present a compelling case for the presence of relativistic reconnection in laser-induced reconnecting plasmas of a given intensity. However, this case is based exclusively on simulations, and such predictions only have as much value as the codes they rely on.

The authors have provided most required information about their numerical setup, but as is, it is hard to extricate. They should state explicitly values for critical density and Debye lengths. From my calculations, it appears the authors have used the physical ion/electron mass ratio, but they should confirm.

Also, it appears from my calculations that they have significantly under resolved the Debye length for the primary simulation in the 40x20x20 box. Why haven't they seen aliasing (self-heating) instabilities in this case? Are they using high-order particle shapes? Smoothing? They need to provide as much information as possible so that other practitioners can reproduce the same results, if so desired.

Reviewer #2 (Remarks to the Author):

The manuscript by Longqing Yi et al presents 3D particle-in-cell (PIC) simulations of magnetic reconnection driven by intense laser-plasma interactions. The authors devise a special target configuration for the study of reconnection, consisting of a thin (1 micron wide) slab followed by an exponential density profile. Fast electron currents produced by the laser and associated return currents along the surface of the target produce strong magnetic fields that can reconnect when the density drops to a level where the fast current can no longer be compensated by the background plasma.

The simulation results are interesting and are likely to trigger further developments in terms of experimental platforms for the study of reconnection, in particular in the relativistic regime, which is still poorly understood. However, it should be noted that the target considered in the simulations is highly idealized and it is far from clear what the impact of more realistic experimental conditions would be on this scheme. The claims that this scheme can be "straightforwardly implemented in experiments" are not justified and need to be carefully revised. Moreover, the claim that "the proposed scenario can significantly improve understanding of fundamental questions" also needs to be addressed. What have we learned in terms of the fundamental behavior of reconnection in this simulation study that we did not know before? And what would be the experimental diagnostics associated with this scheme that would allow us to gain that understanding? This is not clear.

In its current form, this is an interesting simulation work that discusses a highly idealized scheme to study magnetic reconnection, but does not provide much information on the ability to probe the same physics in realistic experimental scenarios and does not provide new insights on the physics of magnetic reconnection. Thus, I cannot recommend its publication in Nature Communications. I suggest that the paper is carefully revised based on the more detailed comments presented below and submitted to a more specialized plasma journal, such as Physics of Plasmas or Physical Review E.

1) The authors discuss some of the features usually associated with reconnection, like the quadruple magnetic field structure and the electron pressure tensor gyrotropy, but do not explain what causes the onset of reconnection in this system. This is a driven system. What is the speed at which field lines

are driven towards each other? If Alfvénic or super-Alfvénic, magnetic flux pile up needs to occur before the conditions for reconnection are established and reconnection can proceed at ~ 0.1 of the local Alfvén speed. It would be useful if the authors could discuss this in light of their simulation results;

2) This manuscript just discusses the electron physics but gives the size of the system in terms of ion inertial lengths. What is the role of the ions in this scheme? What determines the ion dissipation region and what happens there? How does the Alfvén time compare with the duration of the reconnection process? Also, while the electrons can be relativistic, the ions will certainly be non-relativistic. This should be discussed;

3) It is not clear from Figure 4 c) and associated description if the electron spectrum is plotted only for electrons accelerated by reconnection or for all the electrons. If it is for all electrons, how do we distinguish reconnection electrons from laser-driven electrons? If it is only reconnection electrons how were those selected? and where is the thermal population? Reconnection leads to both heating and acceleration, so one should have a thermal component plus a non-thermal component;

4) Short-pulse high-intensity laser-solid experiments are quite complex to perform and diagnose. The authors claim that the proposed scheme allows the study of fundamental questions in magnetic reconnection, but do not discuss what would be the experimental diagnostics that would provide that or what would be learned from such study. The plasma in the reconnection region is at near critical density, so it cannot be probed with conventional optical diagnostics. The scale of the magnetic fields is of the order of microns and varies in fs time scales, which would also be extremely difficult to probe. The electrons accelerated by reconnection would likely move towards the main target. How would they be distinguished from all the other fast electrons produced in the laser-plasma interaction? If the total time-averaged electron spectrum is plotted, can we clearly distinguish between laser-produced electrons and reconnection driven electrons? Also, how do we distinguish the spectrum of reconnection electrons from the spectrum of the return current (which is also relativistic) near the critical density?

5) The authors claim “the proposed scenario can be straightforwardly implemented in experiments.” This is hard to understand given that this paper does not address the implementation of this scheme. Actually, it seems to me that this would be extremely hard to implement experimentally, and it is not even clear it would work under realistic experimental constraints. Just to mention some of the most important aspects:

a) Pointing stability of high-power lasers can be quite poor, meaning that for most shots the laser would not hit the target at the center, but would rather lead to a highly asymmetric interaction. How would this affect the results given the very narrow target of 1 micron?

b) Laser pre-pulse would likely lead to target transverse expansion. This means that in reality one would not have a sharp target-vacuum interface. This should affect the generation of strong surface currents, which is critical for this scheme. How would the results change for a smoother target profile?

c) The plasma density used in the simulations is $20 n_c$ (where n_c is the critical density for the laser propagation). To my knowledge such targets do not exist. How would the results change with more realistic targets with $100s n_c$? This can be addressed in 2D simulations at least in terms of the impact on magnetic field generation on the rear side of the target (where reconnection would occur in reality but not in 2D);

d) Perhaps the most important aspect is that the plasma profile used at the rear side has an

exponential profile in x but retains a sharp, 1 micron wide, profile on y. This is clearly not realistic as the pre-expansion of the target would also be done transversely and not only longitudinally. The authors should provide simulation results with a similar exponential profile in the transverse (y) direction to show how would that affect the interaction and the reconnection process. I suspect that it could significantly impact the ability to generate surface currents and strong magnetic fields, and thus the ability to excite relativistic magnetic reconnection;

Clearly, the implementation of this scheme is not straightforward and the authors should address these different points to support their claims.

6) The authors claim that: "By irradiating a micro-sized plasma slab with a high-intensity laser pulse, it is possible to deposit the laser energy in a very small volume, hence significantly reducing the laser energy (by at least 1-2 orders of magnitude comparing with the results reported in Ref.31)." This needs be clarified. Ref. 31 shows results from two laser systems. A 40 fs short pulse system and a 20 ps long pulse system. The short pulse system is not very different from the one proposed in this manuscript, it uses 2 J in 40 fs with an intensity of $2 \times 10^{19} \text{ W/cm}^2$. This manuscript proposes a 0.25 J system with 50 fs and intensity of 10^{19} W/cm^2 . The main difference seems to come from the higher intensity used in Ref. 31 (which is associated with how relativistic the interaction is) and spot size, which in practice is often limited by the laser system itself. Nothing seems to point towards a specific limitation/advantage of one scheme over the other in terms of laser energy. This claim does not seem fair and should be revised;

7) The claim that this scheme is considerably less expensive to model when compared with previous laser-plasma experiments needs to be revised. Experiments are done at real plasma solid densities of 100s n_c , whereas the simulations of this manuscript were done for unrealistically low densities of 20 n_c . This is where large part of the computational gains arise. Again, this claim does not seem fair.

Reviewer #3 (Remarks to the Author):

The major claims of the paper is that they propose a new experimental geometry where laser-plasmas drive electron beams which interact and drive a relativistic magnetic reconnection. A laser interacting with a thin solid target produces two relativistic electron beams. Overall, the use of a short pulse laser to drive such beams is reasonably well known, and the numerics seem well implemented, so the overall picture and simulation results seem robust. The numerics are well explained and seem like they could be reproduced by other researchers. I might imagine that successfully implementing this in the lab would be an extreme challenge given the small targets and tight tolerances needed. Nonetheless it is a stimulating idea for the field and I found the paper well written and presented.

I do have a few comments which would be important to address-

In contact to this, while the overall acceleration process seems to be well measured, but I have some questions about the details and whether it is appropriate to call it a reconnection acceleration (which would have some astrophysical analog). Here is the devil's advocate question - in this model the reconnection takes place in the presence of extreme 3-d effects, such as the density gradient off the back of the target which allows the beams to interact with each other. Meanwhile, most standard reconnection simulations have reconnection in a system that is quasi-2d (at least in initial condition, see e.g. Guo PRL 2014, Zenitani 2001). Here is one question, are the electrons really being accelerated by electric fields associated with reconnection ($\partial B/\partial t$), or the large electrostatic fields ($\text{grad } \phi$) off the back of the target? Is the role of the magnetic reconnection to demagnetize the

particles to let them experience an electrostatic acceleration? Or is it really a reconnection acceleration?

In contact to this, it would be valuable for the authors to present and discuss some type of simple null test which shows that features of the acceleration depend on the reconnection of magnetic fields, and not the other large fields in the model (such as the large quoted electrostatic fields). I imagine they did this already but it certainly warrants a few sentences (did I miss them?).

The authors discuss Hall reconnection, where there is an important decoupling of the ion and electron flows at the ion skin depth scale. However, the role of the ions is not at all discussed. What role do they play to drive reconnection in this model? My impression is that the ions would be almost motionless on these time and space scales, in which case this is not an appropriate calculation.

A smaller point - My own calculation ($n \sim 10^{27}$, $B = 100$ MG) gives a $\sigma \sim 1$, much less than $\sigma \sim 50$ claimed in the paper. Can the authors elaborate on the conditions that allowed a $\sigma \sim 50$?

Response to Reviewer 1

We are grateful to the referee for carefully reading the manuscript, for the positive remarks and constructive suggestions. As a response to the comments, we note:

The authors present a compelling case for the presence of relativistic reconnection in laser-induced reconnecting plasmas of a given intensity. However, this case is based exclusively on simulations, and such predictions only have as much value as the codes they rely on.

The authors have provided most required information about their numerical setup, but as is, it is hard to extricate. They should state explicitly values for critical density and Debye lengths. From my calculations, it appears the authors have used the physical ion/electron mass ratio, but they should confirm.

Reply: The values for critical density and Debye length are $n_c = 1.1 \times 10^{21} \text{cm}^{-3}$ and $\lambda_D = 1.6 \times 10^{-7} \text{cm}$. The physical ion (proton)/electron mass ratio $m_i/m_e = 1836$ is used in the simulations. We have added this information to the manuscript.

Also, it appears from my calculations that they have significantly under resolved the Debye length for the primary simulation in the 40x20x20 box. Why haven't they seen aliasing (self-heating) instabilities in this case? Are they using high-order particle shapes? Smoothing? They need to provide as much information as possible so that other practitioners can reproduce the same results, if so desired.

Reply: The Debye length not being resolved is rather common in laser-solid PIC simulations, especially in 3D. In the code we are using (EPOCH), the problem of numerical self-heating instabilities is handled by using a higher order particle shape function as well as current smoothing. A detailed discussion can be found in [T. D. Arber, *et. al. Plasma Phys. Control. Fusion* **57** (2015), Sec. 5.1]. The information is now added in the **Methods**. Note the ratio of Debye length and grid space for the $40 \times 20 \times 20$ simulation is almost the same as the example presented in Fig. 4(b) of the Arber PPCF (2015) paper. Moreover, as we mentioned in the **Methods** section of our manuscript, the numerical convergence has been verified by using two different grid steps ($dx = dy = 0.025\lambda_0$ and $0.05\lambda_0$).

Thank you again for the comments! We hope that with our clarifications the paper will be acceptable for publication.

Response to Reviewer 2

We appreciate the careful reading our manuscript and suggestions of improvements. Regarding your comments, we note:

The manuscript by Longqing Yi et al presents 3D particle-in-cell (PIC) simulations of magnetic reconnection driven by intense laser-plasma interactions. The authors devise a special target configuration for the study of reconnection, consisting of a thin (1 micron wide) slab followed by an exponential density profile. Fast electron currents produced by the laser and associated return currents along the surface of the target produce strong magnetic fields that can reconnect when the density drops to a level where the fast current can no longer be compensated by the background plasma.

The simulation results are interesting and are likely to trigger further developments in terms of experimental platforms for the study of reconnection, in particular in the relativistic regime, which is still poorly understood. However, it should be noted that the target considered in the simulations is highly idealized and it is far from clear what the impact of more realistic experimental conditions would be on this scheme. The claims that this scheme can be “straightforwardly implemented in experiments” are not justified and need to be carefully revised. Moreover, the claim that “the proposed scenario can significantly improve understanding of fundamental questions” also needs to be addressed. What have we learned in terms of the fundamental behavior of reconnection in this simulation study that we did not know before? And what would be the experimental diagnostics associated with this scheme that would allow us to gain that understanding? This is not clear.

Reply: We would like to emphasize that **the important novelty of this work is to provide a scheme that can access relativistic reconnection in the low-beta regime.** We have detailed its experimental accessibility above. As shown in Fig. 1, most relativistic MR processes in nature happen in low-beta plasmas (due to strong magnetic fields in most cases), therefore it is of wide interest to explore this poorly understood regime. Although the well-known Biermann-battery-effect based laser-plasma setup can achieve relativistic MR [A. Raymond, *et. al. arXiv:1610.06866* (2016)], it cannot access the low-beta regime. This is due to the fact that the magnetic field lines are driven together by thermal flows, which is critical for the scheme. Our work provides a promising alternative to access this regime **for the first time.**

Fig. 1 Diagram of various reconnection plasmas from lab (green), solar system (red), galaxy (blue) and extra-galaxy (black), the area marked by light blue color indicates relativistic reconnection in the magnetically dominated regime ($\sigma > 1, \beta < 1$); the data used to produce this diagram is adapted from Table.1 in [H. Ji and W. Daughton, *Phys. Plasma* **18**, (2011)]. As we are interested in relativistic MR here, the laser-plasma dot (upper right corner) is updated with the values given in [A. Raymond, *et. al. arXiv:1610.06866* (2016)], and the yellow star shows the parameter of the laser-slab scheme proposed in our manuscript.

We have rephrased the statement “significantly improve understanding of fundamental questions” in the abstract as you suggested. We intended to indicate that by proposing a promising experimental scenario, our work connects state-of-the-art laser plasma experimental capabilities with numerous theoretical and numerical studies on relativistic MR (Ref.[13-18] in our manuscript for instance), many of which are still open questions or under debate. While we are confident that new experiments in the above mentioned regime will trigger further developments in the fundamental knowledge of reconnection, being able to test these theories in laser-plasma experiments is already a significant improvement compared with the current state-of-the-art.

Our manuscript introduces and describes a new experimental platform as well as performs wide-ranging numerical diagnostics (quadrupole fields, relativistic jets, magnetic energy dissipation, particle acceleration mechanism, and so on). To further explore fundamental behavior of reconnection is out of scope of the current paper and will be addressed in future studies.

In its current form, this is an interesting simulation work that discusses a highly idealized scheme to study magnetic reconnection, but does not provide much information on the ability to probe the same physics in realistic experimental scenarios and does not provide new insights on the physics of magnetic reconnection. Thus, I cannot recommend its publication in *Nature Communications*. I suggest that the paper is carefully revised based on the more detailed comments presented below and submitted to a more specialized plasma journal, such as *Physics of Plasmas* or *Physical Review E*.

Reply: We respectfully disagree with the doubt that the same physics cannot be probed in realistic experimental scenarios. We presented in our manuscript an idealized symmetric target configuration for two reasons:

(1) Such a configuration has numerous advantages, such as (i) a length (in x direction) similar to the laser Rayleigh range that allows the laser to pick up as many electrons as possible, therefore generating two high-current beams on each side; (ii) a thickness (in y direction) of the order of 1 μm ensuring that the connected field lines are significantly bent, therefore giving rise to a strong magnetic tension force; (iii) the corona region with a density decreasing ramp mitigates the target normal electrostatic field on the rear side of the slab.

(2) Our work is the first proposal of a new kind of experimental setup, therefore we show a representative case which highlights the key mechanism of the proposed scenario. The physics with an asymmetric setup is worth investigating in detail, but we think a full study should be left to future work.

However, the underlying physical process, **the interaction of two laser-driven electron beams triggering relativistic magnetic reconnection (MR)**, is a very robust mechanism and neither relies on the symmetric configuration nor the above mentioned advantages. While, the laser-slab setup is ideal to study low-beta relativistic MR in the laboratory, in order to implement the idea straightforwardly, one can easily adjust the setup to adapt to current experimental capabilities. To demonstrate this we show an example based on a wire target instead of a slab. Such target can easily be produced, and has already been used in previous experiments, see for example [M. Borghesi, *et. al. Phys. Plasmas* **9**, (2002)]. The corona is an underdense plasma provided by a gas jet and the two electron beams are driven by two laser pulses irradiating the wire on each side. The

setup is illustrated in Fig. 2. As one will see, although the target configuration is changed, the underlying physical process of generating surface current/strong magnetic fields and triggering reconnection is the same, which produces very similar observables (electron jets angular distribution and spectrum), thus demonstrate the robustness of our scheme.

By adjusting the temporal delay (t_d) of two laser pulses, three different simulations are performed for $t_d = 0$, $t_d = 0.5\tau$, and $t_d = \tau$, respectively, where τ is the full duration of laser pulse. Note that the interaction of two laser-driven beams, and therefore MR, can only happen when $t_d < \tau$. The backward propagating electrons with kinetic energy $E_k > 4\text{MeV}$ are proposed as an experimental observable, as they can easily penetrate the wire target and be detected.

Fig. 2 (a) 3D sketch of the laser-wire MR setup: two laser pulses (red arrows) irradiate a wire on each side and drive two electron beams which triggers MR when they interact at the back side of the wire. The whole space (simulation box) is filled with underdense plasma (light blue). The simulation parameters are marked in the figure. The dark blue cylinder in the middle is a solid density wire. (b) Cross section in xy plane, where the laser-driven electrons and reconnection site are marked by green arrows and a yellow cross, respectively. (c) The coordinate system used in the simulation and the definition of the angles θ and ϕ , where v_e is the electron velocity, θ is the angle between v_e and $+x$ direction, and ϕ is the angle between the projection of v_e in the transverse plane (yz) with $+y$ direction.

The angular distribution of these electrons are shown in Fig. 3. There is a significant boost of

the backward propagating electron energy and number when two laser beams temporally overlap, which allows the electron beams to interact with each other and trigger relativistic MR in the middle. The angular distribution pattern in the standard case [Fig. 3(a)] is extended in $\pm z$ direction ($\phi = 90^\circ$ and 270°) due to the magnetic tension force. The pattern in Fig. 3(b) is also leaning towards the $-y$ direction ($\phi = 180^\circ$), because the interaction of two beams is off axis. Figure 3(c) show a dramatic decrease of the backward propagating electrons when the temporal delay of two beams is greater than the duration, which indicates the observed feature is a signature of relativistic MR. Finally, it should be noted the reflection of the laser on the front side of the target will also produce backward propagating electrons, but they are emitted along $\pm y$ direction [shown by the bright spot around $\phi = 0^\circ$ and 180° in Fig. 3 (b-c)], which can be easily separated in experiments. In Fig. 3(d), we show the energy spectrum of the electrons which can be detected with $20^\circ < \phi < 160^\circ$, and $200^\circ < \phi < 340^\circ$ in the three cases.

Fig. 3 Angular distribution of backward propagating ($P_x < 0$) electrons with kinetic energy $E_k > 4\text{MeV}$, for different temporal delays between the two laser pulses (a) $t_d = 0$, (b) $t_d = 0.5\tau$, and (c) $t_d = \tau$. The insets in the bottom right of (a-c) show the corresponding sketch of the setup (cross-section in x - y plane). (d) The energy spectrum of electrons emitted with $20^\circ < \phi < 160^\circ$ and $200^\circ < \phi < 340^\circ$ in three cases. The angles θ and ϕ

are defined in Fig. 2(c).

Details of the setup and the simulation parameters:

1. **Solid Target:** A wire target, with a diameter of $8\ \mu\text{m}$, is used instead of a plasma slab. Although the shape of the target has been changed completely, the mechanism of laser driven relativistic beams and corresponding magnetic fields are the same. The simulation results (in Fig. 3) show emission of similar electron jets [propagating backwards ($-x$) and towards $\pm z$] with slightly smaller cutoff energy ($\sim 8\ \text{MeV}$) than the previous setup. This demonstrates that the scheme is very robust.
2. **Laser:** Two laser pulses irradiate the wire from one side (front side), driving two electron beams which meet on the other side (back side). Synchronization of two lasers may be challenging, but it has been demonstrated before, for example in [B. Aurand, *et. al. Phys. Plasmas* **23**, (2016)]. Moreover, if the laser spot size is larger than the diameter of the wire, one laser beam would be enough. Here we show the two laser case because it gives us the degree of freedom (temporal delay) to perform some simple null tests (as shown in Fig. 3c).
3. **Corona:** The corona is the underdense plasma between two laser-driven electron beams on the back side of the wire. It does not need to have an exponentially decreasing density profile. In the simulation presented in Fig. 2, it is a uniform gas with density $0.01n_c$, which fills the whole simulation box. Such underdense plasma may be provided by a gas jet, which has an exponential spatial distribution. However, for simplicity here we model it with a uniform density as the dimension of the simulation box is much smaller than the scale length of typical gas jet density distribution.
4. **Simulation parameters:** Each laser pulse has the same parameters as the one used in the manuscript. The full incident angle between two lasers is 10° . The initial density of the wire is $20n_c$ (uniform), the diameter is $8\lambda_0$ (in x-y plane), the height is $28\lambda_0$ (in z direction), the background plasma density is $0.01n_c$, the plasma is set to be pre-ionized and the physical ion/electron mass ratio is used. The size of the simulation box is $x \times y \times z = 20\lambda_0 \times 30\lambda_0 \times 30\lambda_0$, which is sampled by $400 \times 600 \times 600$ cells.

A more detailed description of the laser-wire setup will be discussed in a future publication. Nevertheless, the results we presented here demonstrate that the scheme is robust and would work

under realistic experimental constraints.

1) The authors discuss some of the features usually associated with reconnection, like the quadruple magnetic field structure and the electron pressure tensor gyrotropy, but do not explain what causes the onset of reconnection in this system. This is a driven system. What is the speed at which field lines are driven towards each other? If Alfvénic or super-Alfvénic, magnetic flux pile up needs to occur before the conditions for reconnection are established and reconnection can proceed at ~ 0.1 of the local Alfvén speed. It would be useful if the authors could discuss this in light of their simulation results;

Reply: The onset of the reconnection is driven by the combined effect of laser transverse ponderomotive force and Ampère's force between the two electron-beam currents. As the reconnection occurs on a femtosecond time scale, only electrons could react fast enough, and therefore the outflow velocity is (very close to) the speed of light. We did not observe significant magnetic flux pile up in the simulation. By calculating the magnetic flux (Ψ) at the plane at $z=0$ for $x = 25\lambda_0$ to $27\lambda_0$ and simulation time $t = 32T_0$ to $34T_0$ (which is when and where we observe the pressure tensor agyrotropy peaks as shown in Fig. 2 in our manuscript), the average reconnection rate ($d\Psi/dt$) is $0.32cB_0$, with a maximum of $0.47cB_0$ at $t = 33.4T_0$.

A discussion about reconnection rate has now been added in the manuscript.

2) This manuscript just discusses the electron physics but gives the size of the system in terms of ion inertial lengths. What is the role of the ions in this scheme? What determines the ion dissipation region and what happens there? How does the Alfvén time compare with the duration of the reconnection process? Also, while the electrons can be relativistic, the ions will certainly be non-relativistic. This should be discussed;

Reply: The system size is given in terms of laser wavelength. The ions are almost immobile in the time and space discussed in the manuscript, which is the reason why did not mention the physics of ions.

3) It is not clear from Figure 4 c) and associated description if the electron spectrum is plotted only for electrons accelerated by reconnection or for all the electrons. If it is for all electrons, how do we distinguish reconnection electrons from laser-driven electrons? If it is only reconnection

electrons how were those selected? and where is the thermal population? Reconnection leads to both heating and acceleration, so one should have a thermal component plus a non-thermal component;

Reply: Fig. 4(c) in the manuscript shows the electrons initially located in the corona ($x > 21\lambda_0$). They are heated both by reconnection and by the laser pulse. It is impossible to distinguish an electron that is heated by laser or by reconnection. However, the comparison simulation presented in the Supplementary Material highlights the effect of reconnection. In that simulation we removed one of the electron beams by blocking half of the laser pulse on one side, but keeping the energy delivered to the corona approximately the same so that laser heating effect will be similar but reconnection is absent. As one can see from Fig. 1(d) in the Supplementary Material, the heating due to reconnection is significant. Moreover, the reconnection accounts for the backward acceleration which mostly constitute the non-thermal tail, as shown in Fig. 4 below. The thermal population for the original two-beam case at $t = 36T_0$ is approximately 800 keV.

Fig. 4 Corona electron energy spectrum at simulation time $t = 36T_0$ for two-beam (blue curve, primary simulation in the manuscript) and one-beam (red, comparison simulation in the Supplementary Material) case. The black dashed line shows the thermal fit for low energy part of two-beam case.

We have clarified the electrons shown in Fig. 4(c) in the revised manuscript, and the above figure is added as an inset to Fig. 1(d) in the Supplementary Material.

4) Short-pulse high-intensity laser-solid experiments are quite complex to perform and diagnose. The authors claim that the proposed scheme allows the study of fundamental questions in magnetic reconnection, but do not discuss what would be the experimental diagnostics that would provide that or what would be learned from such study. The plasma in the reconnection region is at near critical density, so it cannot be probed with conventional optical diagnostics. The scale of the magnetic fields is of the order of microns and varies in fs time scales, which would also be extremely difficult to probe. The electrons accelerated by reconnection would likely move towards the main target. How would they be distinguished from all the other fast electrons produced in the laser-plasma interaction? If the total time-averaged electron spectrum is plotted, can we clearly distinguish between laser-produced electrons and reconnection driven electrons? Also, how do we distinguish the spectrum of reconnection electrons from the spectrum of the return current (which is also relativistic) near the critical density?

Reply: The relativistic electrons generated have a cut-off energy ~ 10 MeV, and a total charge 0.2nC. They can easily penetrate the main target, and be distinguished from all the other fast electrons by their different energy range and angular distribution. For example, Fig. 3(a-b) shows a clear signature of reconnection, and Fig. 3(c) show fast electrons propagating backwards produced by laser plasma interaction. The differences in electron energy spectrum can be seen in Fig. 3(d). These electrons provide the basic means of diagnostics to study the reconnection.

Moreover, as we have shown in the example above, it is not necessary to drive reconnection in near critical density plasma. If one employs underdense gas plasma as the corona, it is possible to probe it with conventional optical diagnostics. In a near critical density plasma, techniques such as proton imaging or high frequency optics (such as high harmonics and FEL) may provide alternatives. These diagnostics will provide more detailed information about the reconnection region. Although it is difficult to probe magnetic fields of the order of microns in fs time-scale, we have reason to believe these challenges can be overcome in experiments in the near future, for instance, similar experiments have been done to probe micron scale electron microbunches with 2 fs resolution [D. Xiang, *et. al. Phys. Rev. Letts.* **113**, (2014)].

5) The authors claim “the proposed scenario can be straightforwardly implemented in experiments.” This is hard to understand given that this paper does not address the implementation of this scheme. Actually, it seems to me that this would be extremely hard to implement experimentally, and it is

not even clear it would work under realistic experimental constraints. Just to mention some of the most important aspects:

a) Pointing stability of high-power lasers can be quite poor, meaning that for most shots the laser would not hit the target at the center, but would rather lead to a highly asymmetric interaction. How would this affect the results given the very narrow target of 1 micron?

Reply: As our work is the first proposal of driving relativistic MR in low-beta environments based on laser-driven electron beams, we present a symmetric setup in the manuscript as a representative case. But this does not mean the MR process must rely on a highly symmetric configuration. Asymmetry due to laser pointing stability may cause the electron beam currents (and therefore the magnetic fields generated by them) to be different on each side of the slab, but MR will still occur. As an example, in Fig. 3(b), the temporal delay causes a similar asymmetric configuration of electron beams on each side (a weaker current produced by the head of laser pulse on the $-y$ side interacting with a stronger current produced by the peak of laser pulse on the $+y$ side), but one can still observe a clear signature of MR. Therefore, the minimum requirement on pointing stability is that it must be smaller than the spot-size of laser, which can be easily done.

Asymmetric relativistic MR is indeed worth investigating in detail, such as [J. Y. Zhong *et al.*, *Nat. Phys.* **6**, (2010)] did in non-relativistic MR, and we believe our scheme provides a promising experimental possibility. As far as I know, the state-of-the-art high intense laser system can achieve pointing stability $\sim 1 \mu\text{m}$, which may even allow one to control the laser energy on each side of the micro-slab. However, a full study of the asymmetric case should be left to future work.

b) Laser pre-pulse would likely lead to target transverse expansion. This means that in reality one would not have a sharp target-vacuum interface. This should affect the generation of strong surface currents, which is critical for this scheme. How would the results change for a smoother target profile?

c) The plasma density used in the simulations is $20 n_c$ (where n_c is the critical density for the laser propagation). To my knowledge such targets do not exist. How would the results change with more realistic targets with $100s n_c$? This can be addressed in 2D simulations at least in terms of the impact on magnetic field generation on the rear side of the target (where reconnection would occur in reality but not in 2D);

Reply: To address b) and c), we conducted a 2D simulation with density equals to $100n_c$ and has a smooth transverse target profile. The simulation box is $x \times y = 30\lambda_0 \times 15\lambda_0$, sampled by 3000×1500 cells, respectively, the particle per cell is 50 for both ion and electron species. The transverse expansion profile is : $n = n_0 \exp(-(y - y_0)^2/2\sigma_y^2)$, where n_0 is the on-axis density at $y=0$, is the on-axis density at $y=0$, $y_0 = 0.25\lambda_0$, and we show 2 cases with $\sigma_y = 0.25\lambda_0$ and $0.5\lambda_0$ in Fig. 5(a-b) and Fig. 5(c-d), respectively.

Fig. 5 The initial density profile and static magnetic fields generated in 2D simulations (a-b) $n_0 = 100n_c$, transverse scale length $\sigma_y = 0.25\lambda_0$; (c-d) $n_0 = 100n_c$, transverse scale length $\sigma_y = 0.5\lambda_0$; (e-f) show the primary 3D simulation we used in the manuscript (2D cut at plane $z=0$), with $n_0 = 20n_c$ and sharp transverse boundary.

The figure shows that a smooth transverse profile and realistic $100n_c$ density does not significantly change the surface current and magnetic field generation process. The static magnetic fields in Fig. 5(d) are about half the value in the 3D simulation with sharp boundary [Fig. 5(f)] but this is mainly due to the increase of the effective slab thickness (defined as the transverse distance

between two critical density layers), rather than due to the realistic $100n_c$ density or smooth transverse profile.

Therefore, as long as the transverse expansion of the slab due to the laser pre-pulse is much smaller than the laser spot, it does not change the results. Nevertheless, even if this condition is not satisfied, one could split the laser pulse to shoot on each side of the slab similar to the laser-wire setup we show in Fig. 2.

d) Perhaps the most important aspect is that the plasma profile used at the rear side has an exponential profile in x but retains a sharp, 1 micron wide, profile on y . This is clearly not realistic as the pre-expansion of the target would also be done transversely and not only longitudinally. The authors should provide simulation results with a similar exponential profile in the transverse (y) direction to show how would that affect the interaction and the reconnection process. I suspect that it could significantly impact the ability to generate surface currents and strong magnetic fields, and thus the ability to excite relativistic magnetic reconnection;

Reply: First of all, as we have shown in Fig. 2-3, the corona can be a gas plasma, which would also work for the slab target. This can be one of the solutions to the concern that was raised in d). When it comes to the original configuration, note that the longitudinal scale length of the density-ramp in the x direction that is used in our manuscript ($2\lambda_0$) could be realized by ablating the rear side of the slab with a long laser pulse. A similar density-ramp in the transverse direction would only exist on the rear side, therefore it should not have a significant impact on the ability to generate the surface current and strong magnetic fields, which mainly happens on the front side. In addition, the pre-expansion of the rear target in the transverse direction would neither be an obstacle to the propagation of laser-driven electrons (with energies ~ 30 MeV), nor will it prevent the two electron beams interacting with each other [only the plasma in between matters (which has been considered in our manuscript)].

Clearly, the implementation of this scheme is not straightforward and the authors should address these different points to support their claims.

Reply: A few brief comments have been added in the manuscript regarding the issues mentioned above, and we anticipate the material of the reply will be included as supplementary material to support our claims.

6) The authors claim that: “By irradiating a micro-sized plasma slab with a high-intensity laser pulse, it is possible to deposit the laser energy in a very small volume, hence significantly reducing the laser energy (by at least 1-2 orders of magnitude comparing with the results reported in Ref.31).” This needs be clarified. Ref. 31 shows results from two laser systems. A 40 fs short pulse system and a 20 ps long pulse system. The short pulse system is not very different from the one proposed in this manuscript, it uses 2 J in 40 fs with an intensity of 2×10^{19} W/cm². This manuscript proposes a 0.25 J system with 50 fs and intensity of 10^{19} W/cm². The main difference seems to come from the higher intensity used in Ref. 31 (which is associated with how relativistic the interaction is) and spot size, which in practice is often limited by the laser system itself. Nothing seems to point towards a specific limitation/advantage of one scheme over the other in terms of laser energy. This claim does not seem fair and should be revised;

Reply: The plasma beta $\beta \propto nT/B^2$ is the ratio of thermal energy and magnetic energy density. The Biermann-battery-effect based laser-plasma scheme (such as Ref. [31]), operates in the high beta regime ($\beta = 10 - 100$), while our scenario works in $\beta < 1$. This means that in order to achieve the same reconnection magnetic field, the total energy density required in the previous Biermann battery effect scheme is 1-2 orders of magnitude larger than our scheme, just to heat the plasma. To calculate the total energy, one needs to multiply the energy density with the volume. In our scheme, the system size is on the order of 1-10 μm , while the other scheme is 10-100 μm (take Ref.[31] for example). This is one of the advantages of our scheme in terms of laser energy. This point has been clarified in the revised manuscript.

7) The claim that this scheme is considerably less expensive to model when compared with previous laser-plasma experiments needs to be revised. Experiments are done at real plasma solid densities of $100s n_c$, whereas the simulations of this manuscript were done for unrealistically low densities of $20 n_c$. This is where large part of the computational gains arise. Again, this claim does not seem fair.

Reply: This claim has been removed in the revised version of the paper.

Thank you again for all your comments! We hope that with these amendments and clarifications our manuscript is acceptable for publication.

Response to Reviewer 3

We appreciate the careful reading our manuscript, your positive remarks and suggestions of improvements. Regarding your comments, we note:

The major claims of the paper is that they propose a new experimental geometry where laser-plasmas drive electron beams which interact and drive a relativistic magnetic reconnection. A laser interacting with a thin solid target produces two relativistic electron beams. Overall, the use of a short pulse laser to drive such beams is reasonably well known, and the numerics seem well implemented, so the overall picture and simulation results seem robust. The numerics are well explained and seem like they could be reproduced by other researchers. I might imagine that successfully implementing this in the lab would be an extreme challenge given the small targets and tight tolerances needed. Nonetheless it is a stimulating idea for the field and I found the paper well written and presented.

Reply: Thank you for your positive comments. To demonstrate the robustness of the key physical processes, in the Response to Reviewer 2 we show another experimental setup based on a wire target instead of a slab, with a corona that is an underdense plasma provided by a gas jet and the two electron beams are driven by two laser pulses irradiating the wire on each side. This shows that while the laser-slab setup is ideal to study low-beta relativistic MR in the laboratory, in order to implement the idea straightforwardly, one can easily adjust the setup to adapt to current experimental capabilities.

I do have a few comments which would be important to address-

In contact to this, while the overall acceleration process seems to be well measured, but I have some questions about the details and whether it is appropriate to call it a reconnection acceleration (which would have some astrophysical analog). Here is the devil's advocate question - in this model the reconnection takes place in the presence of extreme 3-d effects, such as the density gradient off the back of the target which allows the beams to interact with each other. Meanwhile, most standard reconnection simulations have reconnection in a system that is quasi-2d (at least in initial condition, see e.g. Guo PRL 2014, Zenitani 2001). Here is one question, are the electrons

really being accelerated by electric fields associated with reconnection (dB/dt), or the large electrostatic fields ($\text{grad } \phi$) off the back of the target? Is the role of the magnetic reconnection to demagnetize the particles to let them experience an electrostatic acceleration? Or is it really a reconnection acceleration?

In contact to this, it would be valuable for the authors to present and discuss some type of simple null test which shows that features of the acceleration depend on the reconnection of magnetic fields, and not the other large fields in the model (such as the large quoted electrostatic fields). I imagine they did this already but it certainly warrants a few sentences (did I miss them?).

Reply: The electrons gain energy both from reconnection ($d\Psi/dt$, where Ψ is the magnetic flux) and the electrostatic fields. However, **the contribution from reconnection is significant enough that it will produce a clear signature that can be observed in experiments.** For example, by calculating the reconnection flux change with time in the simulation, one gets a peak reconnection rate $E_r = 0.47cB \approx 2.46 \text{ TV/m}$, compared with the peak electric field $E \approx 5.76 \text{ TV/m}$ in the simulation (total field of reconnection and electrostatic fields), which means about 43% of the acceleration comes from the reconnection field.

The complication that part of the acceleration is due to electrostatic fields exists not only in our scheme, but also in other laser-plasma MR setups. For example, in the simulation presented in [A. Raymond, *et. al. arXiv:1610.06866* (2016)], the authors calculate reconnection field by subtracting a capacitive sheath field from the target normal electric field (see Fig. 5 in their paper).

The setup could be optimized to reduce the part of the acceleration that is caused by electrostatic fields. A possible solution is that instead of using a slab, one can design a wedge target (in x-y cross section), so that the tip side (pointing at reconnection site) does not generate strong target normal electric field in the direction of reconnection acceleration. However, optimization of the scheme in this direction is out of the scope of the present paper and should be left for future work.

We have shown a null test in the **Supplementary Material**. In Fig. 3(c) of the Response to Reviewer 2, we show another one to highlight the experimental signatures. Clear differences can be seen in the angular distribution and energy spectrum of backward propagating electrons with and without the reconnection effect.

The authors discuss Hall reconnection, where there is an important decoupling of the ion and electron flows at the ion skin depth scale. However, the role of the ions is not at all discussed.

What role do they play to drive reconnection in this model? My impression is that the ions would be almost motionless on these time and space scales, in which case this is not an appropriate calculation.

Reply: The ions are almost immobile in the system size we discussed in the manuscript, therefore the electrons and ions are naturally decoupled.

A smaller point - My own calculation ($n \sim 10^{27}$, $B = 100$ MG) gives a $\sigma \sim 1$, much less than $\sigma \sim 50$ claimed in the paper. Can the authors elaborate on the conditions that allowed a $\sigma \sim 50$?

Reply: The boost of σ is due to local density decreasing when large number of electrons are accelerated out of reconnection site. The maximum σ at a cross-section close to the reconnection X-point is about 30, as shown by the figure below. (We gave the value 30 in the paper, not 50.)

Fig. 6 Magnetization parameter at a cross section close to mid-plane (reconnection X-point) at $y = 0.1\lambda_0$.

Thank you again for all your comments! We hope that with these amendments and clarifications our manuscript is acceptable for publication.

Reviewers' comments:

Reviewer #1 (Remarks to the Author):

The authors have satisfactorily addressed my earlier points. The manuscript has been improved, particularly in view of the additional supplementary material. I already thought they had made a compelling case, and this is even more the case in this new version.

Reviewer #2 (Remarks to the Author):

The authors have made a significant effort to show that their scheme is robust to variations in the plasma profiles (e.g. the expanded target and the wire target immersed in low-density gas). This clearly addresses one of the main points I had previously and improves the manuscript. I suggest that the authors add a sentence or two to the manuscript discussing the robustness of the results (as they did in their response) and add some version of Figure 5 of the response to the supplementary material.

As I mentioned in my last report, the simulation results are interesting and their validity is well justified. The main question that remains, in my opinion, is whether this work represents a significant advance for this community and thus deserves publication in Nature Communications. It may indeed deserve, but that is not clear in the way the novelty/importance of the results is justified in the current version.

The argument of the authors is that the main novelty/interest of this work is that their setup uniquely allows probing a low β ($\beta \ll 1$) reconnection regime. However they do not show or argue convincingly why that is indeed the case. In this regard I see two problems with the current version, specified below, that need to be addressed to be convinced that this represents indeed a significant advance.

1) The importance of the low β regime in the context of the proposed setup

The authors invoke the interest in the low β regime in astrophysics, which is clear, but the regime they are studying is very different from astrophysical environments of interest. The magnetic field annihilation in the proposed setup is driven by the fast (non-thermal) electrons produced by the laser and the B-fields associated with their current. This is significantly different from the physics evoked in astrophysical environments. In my opinion, they should clarify this and mention that this is a different but interesting regime that could potentially be studied experimentally (i.e. more focus on interesting physics associated with 3D, strongly driven regime vs simply saying that it has direct applicability to relativistic reconnection in astrophysics.) In the current version of the manuscript, the role that the ambient electron temperature is playing is not discussed. In fact, looking at the spectrum of Fig. 4c, we see that we don't have a colder thermal background plasma plus an energetic non-thermal component. Instead, it looks like a single power-law distribution, with the maximum energy changing in time. In such case, the calculation of a thermal component does not seem well justified. How is a 6 keV temperature calculated from a power-law spectrum? For such a distribution, I believe it is more meaningful to compare the electron energy density with the magnetic energy density. Put in order words, we want to compare the total electron pressure (not just thermal) with the magnetic pressure. Based on Figure 4c, we can see that it changes in time. For instance near 36 T0, I estimate that the average energy density is ~ 40 keV (is this correct?), and thus the ratio of electron energy density to magnetic energy density is ~ 0.16 (instead of 0.02). This discussion is much more important than

simply saying that we want to probe low β without explaining why is β an important parameter in this configuration (where we don't really have an equilibrium and a thermal plasma), and how do the results change with β (or more meaningful in my opinion, with the ratio of electron energy density to magnetic energy density).

2) Uniqueness of proposed setup to probe low β (magnetically dominated) reconnection

The authors put a significant emphasis on the advantage of their scheme over previous schemes (particularly the work by A. Raymond et. al. arXiv:1610.06866 (2016)), in terms of the β parameter achieved. I think it is important to be more careful in the comparison that is being done. The authors simply used the β value obtained in the simulations of the work by A. Raymond et. al. and compare it with their simulations. But I do not see any claim or reason as to why the β value obtained in a different setup is the minimum possible (this was to the subject of the study of Raymond et al.) In other words, it is not clear from the manuscript that the proposed setup really offers an advantage there. Again, noting that the more meaningful number is the ratio of electron energy density to the magnetic energy density, we see that this is basically determined by the laser-plasma interaction in both cases and in particular the electron population (which determines both the electron energy density and the current that drives the B-fields). In the case of an intense laser irradiating a solid foil (as explained by A. Raymond et. al. arXiv:1610.06866 (2016)), the fast electrons that are accelerated along the foil surface carry the magnetic field. As the electrons propagate away from the interaction region, the fields decay, and thus the ratio of electron energy density to magnetic energy density is a function of the distance from the laser interaction where the B-field annihilation occurs. As explained by Raymond et. al., near the laser interaction region the B-field amplitude also reaches 100 MG, which is typical, as also observed in the current manuscript. They do not discuss what is the energy density of the fast electrons confined to the surface that carry the B-field and how it changes with propagation. Based on this, the authors should present a more detailed analysis of why their scheme can uniquely produce conditions that are magnetically dominated to support their claim regarding the importance of this new setup.

Reviewer #3 (Remarks to the Author):

The authors have answered most questions raised. However, a point about the reconnection parameters still needs to be addressed, and a point raised by Referee 2 on what important physics of the relativistic regime could be studied I don't think has been sufficiently presented.

1. The technical points of the particle energization and null test have been adequately addressed; nice work.

2. Second, one of my points has not been adequately addressed, which comes to the Hall and ion physics, and therefore system size.

The authors conclude that the ion's do not play a role, and this seems very reasonable. However, in this case why do they still compare the system size to the ion thermal gyro-radius (and then compare to other recent experiments based on this metric?) (This is the "calculation" I referred to in my previous report as "not appropriate". I apologize that about half of the thought was not included.)

So, first of all, I believe this comparison needs to be removed.

A second point is that the system size is nonetheless still important, and that the authors must make

a more careful job of presenting the system size regime obtained.

Using a value of density reported by the authors (for $n/n_c \sim 0.01$, to obtain the σ they claim), I find that the local d_e in the reconnection region must be $\sim 1.6 \mu\text{m}$. This implies that their whole reconnection volume in z (outflow) $\sim 3 d_e$, inflow $y \sim 2 d_e$, and $x \sim 2 d_e$ (where x is along the reconnection current - a direction usually assumed to be isotropic in most work!).

Such a small L/d_e is very marginal for generating magnetic reconnection. It is basically the minimum scale where the plasma can respond collectively to the magnetic field. (System size matters: as an somewhat facetious counter-point, imagine taking bar magnets ($B \sim 1 \text{ kG}$), to outer space ($n \sim 1 \text{ cm}^{-3}$) and obtain a $\sigma \sim 10^5$! Did I just get enormous σ for free? No: d_e in this case is 0.5 km , so actually I need a giant set of bar magnets... the system size matters.)

In contrast, I calculate for laser experiments, e.g. of Li et al (2007) $L / d_e \sim 5000$ for $n_e \sim 2 \cdot 10^{20} \text{ cm}^{-3}$ and a 2 mm current sheet. Recent state of the art simulations are at $L / d_e \sim$ several hundred (Guo PRL 2014). In astrophysical environments L / d_e will be enormous.

The authors need to (1) state in the manuscript what system size L/d_e is obtained during their relativistic reconnection phase. (Not in terms of an inapplicable ion gyro-radius), and (2) if indeed it is approximately as small as I have estimated, to justify how this small L/d_e is still a valid regime for studying reconnection (e.g. other simulations studying at this scale?).

3. I need to pick up a point raised by Referee 2, which I do not think the present version adequately addresses - what fundamental physics of relativistic magnetic reconnection can be addressed by this scheme?

Much is known about non-relativistic reconnection, including its role to energize particles. What is novel about magnetic reconnection in relativistic plasmas, which can be addressed by this proposed experiment platform? If a relativistic reconnection simply involves a rescaling of parameters into relativistic counterparts, then what is really new? The authors list several theory publications in which relativistic reconnection is studied - but what controversies or novel aspects of the relativistic regime demand experimental verification? Some of the signatures observed (particle acceleration, agyrotropy and Hall fields) are not unique to relativistic MR.

Response to Reviewer 2

We sincerely appreciate the careful reading of our manuscript, the positive remarks and constructive suggestions. As a response to the comments, we note:

The authors have made a significant effort to show that their scheme is robust to variations in the plasma profiles (e.g. the expanded target and the wire target immersed in low-density gas). This clearly addresses one of the main points I had previously and improves the manuscript. I suggest that the authors add a sentence or two to the manuscript discussing the robustness of the results (as they did in their response) and add some version of Figure 5 of the response to the supplementary material.

Reply: We agree that the above mentioned points need to be emphasized. The manuscript and supplementary material have been revised as suggested.

As I mentioned in my last report, the simulation results are interesting and their validity is well justified. The main question that remains, in my opinion, is whether this work represents a significant advance for this community and thus deserves publication in Nature Communications. It may indeed deserve, but that is not clear in the way the novelty/importance of the results is justified in the current version.

The argument of the authors is that the main novelty/interest of this work is that their setup uniquely allows probing a low β ($\beta \ll 1$) reconnection regime. However they do not show or argue convincingly why that is indeed the case. In this regard I see two problems with the current version, specified below, that need to be addressed to be convinced that this represents indeed a significant advance.

1) The importance of the low β regime in the context of the proposed setup

The authors invoke the interest in the low β regime in astrophysics, which is clear, but the regime they are studying is very different from astrophysical environments of interest. The magnetic field annihilation in the proposed setup is driven by the fast (non-thermal) electrons produced by the laser and the B-fields associated with their current. This is significantly different from the physics evoked in astrophysical environments. In my opinion, they should clarify this and mention that this is a different but interesting regime that could potentially be studied experimentally (i.e. more focus on interesting physics associated with 3D, strongly driven regime vs simply saying that it has direct applicability to relativistic reconnection in astrophysics.)

Reply: The introduction part of our manuscript has been revised to address the novelty/importance of the presented results.

We agree that the proposed setup is different from relativistic reconnection in astrophysical environments. In the revised version of the manuscript we put the emphasis on the interesting physics in this new regime that could be studied by our scheme. However, we think the setup could nonetheless shed light on *some* of the physics of relativistic reconnection in astrophysics, for two reasons:

(i) Many of the astrophysical processes of interest do not rely crucially on the onset of reconnection. For instance, the “kinetic beaming” effect [Uzdensky, et. al. *The Astrophys. J. Letts.* **737**, (2011)], where the most energetic particles form compact bunches in the reconnection layer, could be tested by the proposed scheme. Kinetic beaming is a result of the acceleration of relativistic particles in the reconnection electric fields and deflection by the reconnected fields.

(ii) Some of the reconnection processes in astrophysics are also in the strongly driven regime, and our setup could potentially be used to study some of the interesting physics associated with this effect. A possible example is particle acceleration in compression-driven reconnection in the striped pulsar wind [Lyubarsky, et. al. *The Astrophys. J.* **682**, (2008)]. Although the validity of using the attraction of laser-driven electron beams to mimic the compression effect of termination shock may need to be justified in future studies, obtaining a scheme that could accomplish relativistic reconnection in the low- β regime is a promising first step.

The importance of achieving the low β regime in the two examples listed above is clear: if the role played by magnetic reconnection is negligible, in

terms of energy balance, and the energization of particles depends mainly on other mechanisms (i.e. laser heating), the signatures that one looks for may be hidden in a thermal background and therefore be very difficult to diagnose.

In the current version of the manuscript, the role that the ambient electron temperature is playing is not discussed. In fact, looking at the spectrum of Fig. 4c, we see that we don't have a colder thermal background plasma plus an energetic non-thermal component. Instead, it looks like a single power-law distribution, with the maximum energy changing in time. In such case, the calculation of a thermal component does not seem well justified. How is a 6 keV temperature calculated from a power-law spectrum?

Reply: A colder thermal background plasma plus an energetic non-thermal component can not clearly be seen in Fig. 4(c) of our manuscript because laser-plasma-interaction (LPI) gives rise to a two-temperature electron energy distribution (which is common in laser-solid interaction, e.g. [Nishiuchi, et. al. *Phys. Rev. A* **357**, (2006)], [DuBois, et. al. *Phys. Plasmas* **24**, (2017)]). As can be seen in Fig. 1 below (which is Fig. 4 in our previous response, note the longitudinal scale is different from Fig. 4(c)), the 1-beam (no reconnection) and 2-beam (with reconnection) case give similar two-temperature distributions in the low-energy part, which indicates that it results from LPI.

Figure 1: Two-temperature distribution of the coronal electrons at $t = 36T_0$ (at the low-energy side). The blue curve shows the main simulation in our manuscript, and the red shows curve shows the comparison 1-beam simulation in the Supplementary material.

The 6 keV estimate we used in our manuscript is calculated from the lower temperature part at simulation time $30T_0$. It evolves with time, at $t = 36T_0$ (shown in Fig. 1), it is increased to approximately 40 keV. Note this part of the spectrum (electron kinetic energy < 0.2 MeV) contains 98.9% of total charge and 87.7% of total kinetic energy in the corona, so that it gives a rather close estimate of the total electron energy density.

For such a distribution, I believe it is more meaningful to compare the electron energy density with the magnetic energy density. Put in other words, we want to compare the total electron pressure (not just thermal) with the magnetic pressure. Based on Figure 4c, we can see that it changes in time. For instance near $36 T_0$, I estimate that the average energy density is 40 keV (is this correct?), and thus the ratio of electron energy density to magnetic energy density is 0.16 (instead of 0.02). This discussion is much more important than simply saying that we want to probe low β without explaining why is β an important parameter in this configuration (where we don't really have an equilibrium and a thermal plasma), and how do the results change with β (or more meaningful in my opinion, with the ratio of electron energy density to magnetic energy density)

Reply: A discussion on the estimation of β as well as the evolution trend with time has been added in the manuscript.

The importance of magnetically dominated regime ($\beta < 1$) in this configuration is that the magnetic energy released by reconnection accounts for a significant amount of total energy transition (as can be seen in Fig. 4(b) of our manuscript), and this ensures that the high energy tail in the electron spectrum is associated with reconnection (as in Fig. 1). On the other hand, if $\beta \gg 1$, the contribution from magnetic reconnection in the total electron energy density is almost negligible. Then the main contribution to the electron energy increase and probably even the cut-off energy of electrons are determined by another source (LPI in this case). It is therefore more difficult to probe the reconnection associated effects in the $\beta \gg 1$ case. The range of β factor variation in the presented scheme is from 0.07 to 0.22, therefore it does not affect our claims in this regard.

2) Uniqueness of proposed setup to probe low β (magnetically dominated) reconnection

The authors put a significant emphasis on the advantage of their scheme over previous schemes (particularly the work by A. Raymond et. al. arXiv:

1610.06866 (2016)), in terms of the β parameter achieved. I think it is important to be more careful in the comparison that is being done. The authors simply used the β value obtained in the simulations of the work by A. Raymond et. al. and compare it with their simulations.

Reply: The claim that previous laser-plasma reconnection studies (Biermann-battery-effect based) are in the high β regime is not just a simple calculation from the simulation work by A. Raymond et. al., but it is based on vast number of experimental results as summarised in [Fox, et. al. *Phys. Plasmas* **19**, (2012)]. This is a well-accepted characteristics of such experiments, so that in most of the simulation works on laser-plasma reconnection (although they usually do not model the laser-plasma interaction and the formation of the magnetic bubbles), the initial condition is set to be a high- β plasma [Totorica, et. al. *Phys. Plasmas* **24**, (2017)], [Fox, et. al. *Phys. Rev. Lett.* **106**, (2011)], [Huang, et. al. *Phys. Plasmas* **24**, (2017)].

But I do not see any claim or reason as to why the β value obtained in a different setup is the minimum possible (this was to the subject of the study of Raymond et al.) In order words, it is not clear from the manuscript that the proposed setup really offers an advantage there. Again, noting that the more meaningful number is the ratio of electron energy density to the magnetic energy density, we see that this is basically determined by the laser-plasma interaction in both cases and in particular the electron population (which determines both the electron energy density and the current that drives the B-fields). In the case of an intense laser irradiating a solid foil (as explained by A. Raymond et. al. arXiv:1610.06866 (2016)), the fast electrons that are accelerated along the foil surface carry the magnetic field. As the electrons propagate away from the interaction region, the fields decay, and thus the ratio of electron energy density to magnetic energy density is a function of the distance from the laser interaction where the B-field annihilation occurs. As explained by Raymond et. al., near the laser interaction region the B-field amplitude also reaches 100 MG, which is typical, as also observed in the current manuscript. They do not discuss what is the energy density of the fast electrons confined to the surface that carry the B-field and how it changes with propagation. Based on this, the authors should present a more detailed analysis of why their scheme can uniquely produce conditions that are magnetically dominated to support their claim regarding the importance of this new setup.

Reply: In [Raymond, et. al. *arXiv: 1610.06866* (2016)], the authors proposed to use a sub-picosecond (short pulse regime), high intensity ($\sim 10^{19}$

W/cm²) laser in order to reach relativistic reconnection regime. This is relatively different from the previous results reported before, as most of them use nanosecond-duration laser pulses (long-pulse regime). This is a new regime in terms of laser-plasma reconnection study, but the heating of electrons by LPI in this regime has been extensively studied because it is closely related to target normal sheath field acceleration (TNSA).

These TNSA studies have shown that the generation of hot electrons becomes more and more effective when laser intensity increases from $\sim 10^{15}$ W/cm² to $\sim 10^{20}$ W/cm², as can be seen in Fig. 6 of [Macchi, et. al. *Rev. Mod. Phys.* **85**, (2013)]. The energy density of hot electrons can be obtained from energy flux balance: $\eta_h I \approx n_h v_h T_h$, where $\eta_h \approx 10\% \sim 30\%$ is the ratio of laser energy that transfers to hot electrons, I is the laser intensity, n_h , v_h and T_h are the density, velocity, and temperature of hot electrons, respectively [Sec. II.B.2 of Macchi, et. al. *Rev. Mod. Phys.* (2013)].

By plugging in $I = 10^{19}$ W/cm², and $v_h \approx c$, one obtains the electron energy density $n_h T_h \approx 3.3 \times 10^{13} - 1 \times 10^{14}$ J/m³. Note, that this is the energy density for thermal electron distribution after the expansion. The initial electron energy density (non-thermal) near the laser interaction region can be estimated by the average density in the target skin layer (a few n_c) times the ponderomotive energy ($\sim a_0^2 mc^2/2$), which roughly yields 10^{15} J/m³. A more rigorous discussion can be found in [M. Roth, et. al. *CERN Yellow Report* (2016)] (available at <https://e-publishing.cern.ch/index.php/CYR/article/view/222>).

Therefore, for $B \sim 100$ MG at the laser interaction region, even with the expanded electron energy density ($n_h T_h \approx 3.3 \times 10^{13} - 1 \times 10^{14}$ J/m³) gives the plasma $\beta \sim 1 - 3$. Using a reduced $B \sim 35$ MG in the reconnection layer reported in [Raymond, et. al. *arXiv: 1610.06866* (2016)] yields $\beta \sim 10 - 30$, which roughly agrees with their simulation.

It is difficult to make a rigorous analysis of the “minimum possible” β for the previous laser-plasma setup, but it is clear that a high absorption factor η_h is mainly responsible for the high β plasma in that case. Some experiments have shown for a ultra-intense laser $\sim 10^{20}$ W/cm⁻³, the η_h is even higher, and could reach up to 60% [Ping, et. al. *Phys. Rev. Letts* **100**, (2008)], which means that even if all the rest of the laser energy transfers to magnetic energy, β is still greater than one (we assume the volume is the same since the magnetic field moves with the hot electrons).

On the other hand, in our scheme the laser heating is much lower because the laser only passes through the slab, therefore the deposited laser energy is much less. The most significant heating occurs at the very-left

side thus it does not affect the reconnection that happens in the corona. The laser-accelerated surface electrons propagate mostly outside the reconnection layer, as can be seen in Fig. 4(a) of our manuscript.

Thank you again for your comments! We hope that with our clarifications and amendments you can recommend our paper for publication.

Sincerely Yours,
The Authors

Response to Reviewer 3

We sincerely appreciate the careful reading of our manuscript, the positive remarks and constructive suggestions. As a response to the comments, we note:

The authors have answered most questions raised. However, a point about the reconnection parameters still needs to be addressed, and a point raised by Referee 2 on what important physics of the relativistic regime could be studied I don't think has been sufficiently presented.

1. The technical points of the particle energization and null test have been adequately addressed; nice work.
2. Second, one of my points has not been adequately addressed, which comes to the Hall and ion physics, and therefore system size.

The authors conclude that the ion's do not play a role, and this seems very reasonable. However, in this case why do they still compare the system size to the ion thermal gyro-radius (and then compare to other recent experiments based on this metric?) (This is the "calculation" I referred to in my previous report as "not appropriate". I apologize that about half of the thought was not included.)

So, first of all, I believe this comparison needs to be removed.

Reply: We agree, thank you for pointing this out. This comparison has been removed in the revised version of the manuscript.

A second point is that the system size is nonetheless still important, and that the authors must make a more careful job of presenting the system size regime obtained.

Using a value of density reported by the authors (for $n/n_c \sim 0.01$, to obtain the sigma they claim), I find that the local d_e in the reconnection region must be $\sim 1.6\mu m$. This implies that their whole reconnection volume in z (out-flow) $\sim 3d_e$, inflow $y \sim 2d_e$, and $x \sim 2d_e$ (where x is along the reconnection current - a direction usually assumed to be isotropic in most work!).

Such a small L/d_e is very marginal for generating magnetic reconnection. It is basically the minimum scale where the plasma can respond collectively to the magnetic field. (System size matters: as an somewhat facetious counterpoint, imagine taking bar magnets ($B \sim 1$ kG), to outer space ($n \sim 1$ cm $^{-3}$) and obtain a sigma $\sim 10^5$! Did I just get enormous sigma for free? No: d_e in this case is 0.5 km, so actually I need a giant set of bar magnets... the system size matters.)

In contrast, I calculate for laser experiments, e.g. of Li et al (2007) $L/d_e \sim 5000$ for $n_e \sim 2 \times 10^{20}$ cm $^{-3}$ and a 2 mm current sheet. Recent state of the art simulations are at $L/d_e \sim$ several hundred (Guo PRL 2014). In astrophysical environments L/d_e will be enormous.

The authors need to (1) state in the manuscript what system size L/d_e is obtained during their relativistic reconnection phase. (Not in terms of an inapplicable ion gyro-radius), and (2) if indeed it is approximately as small as I have estimated, to justify how this small L/d_e is still a valid regime for studying reconnection (e.g. other simulations studying at this scale?).

Reply: (1) The system size L/d_e is now stated in the manuscript. Note that when we report the *maximum* σ factor to be $\sigma_m \sim 30$, we already ignored the area with local plasma density lower than $0.01n_c$ in order to avoid results that are not meaningful. The *average* plasma density in the reconnection phase is estimated to be $n/n_c \sim 0.1$. This gives the reconnection volume as z (outflow) $\sim 8d_e$, y (inflow) $\sim 8d_e$, and $x \sim 4d_e$ (and $z \sim 25\rho_0$, $y \sim 25\rho_0$, and $x \sim 13\rho_0$, respectively, where $\rho_0 = c/\omega_{ce}$ is the electron gyroradius).

(2) Even if our estimates are somewhat larger than the reviewer's, we agree that this is a fairly small system size. To justify that it is a valid regime for studying reconnection, we can compare to some of the earlier studies in the area:

(i) Most of the previous simulation studies on relativistic reconnection focus on the "multiple X-line" reconnection regime [Ji, et. al. *Phys. Plasmas* **18**, (2011)]. This is associated with large system sizes. However, the

simulations are usually initiated by a pair of Harris-type currents. Although the simulation domains are usually several hundred d_e , the (full) width of the Harris-type current layer are just a few d_e (for instance, $\sim 2.5d_e$ in [Dahlin, et. al. *Phys. Plasmas* **21**, (2014)], $\sim 12d_e$ in [Guo, et. al. *Phys. Rev. Letts.* **113**, (2014)], and $\sim 1.3d_e$ in [Nalewajko, et. al. *The Astrophys. J.* **815**, (2015)]). During the time-scale in their study, many reconnection events happen, as the plasmoids become larger and larger, merge and eventually become comparable to the simulation domain. However, in the beginning of this evolution, the sizes of the plasmoids are just on the order of the current layer thickness (as can be seen for example in Fig. 1 in [Nalewajko, et. al. *The Astrophys. J.* **815**, (2015)]). If one considers two neighbouring plasmoids merging in this stage, the system size is similar to the one in our manuscript. Therefore we believe that our system is an equally valid regime for studying reconnection, as e.g. in [Nalewajko, et. al. *The Astrophys. J.* **815**, (2015)], where particle acceleration by reconnection in this early stage was studied. Fig. 2 in their paper shows reconnection associated electric fields in the X-points at $\omega_c t \sim 250$ (middle row).

(ii) The laser-plasma based reconnection studies are mainly in the “single X-line” regime (as in our manuscript). Comparing with our paper, the system size in these studies, if measured by electron inertial length, appears to be larger because most of them are non-relativistic reconnections. For instance, in [Nilson, et. al. *Phys. Plasmas* **15**, (2008)], the plasma density is $\sim 5 \times 10^{19} \text{ cm}^{-3}$ (Fig. 3 in their paper), and laser spot separation on target $L \sim 400 \mu\text{m}$, which gives $L/d_e \sim 500$. However, since the magnetic field in this case is typically $\sim 1 \text{ MG}$, their system size is $L \sim 20\rho_0$ (in units of electron gyroradius), which is comparable to our case.

In relativistic reconnection, the system size appears to be smaller in terms of electron inertial length only because $d_e/\rho_0 = \sqrt{\sigma}$ (thus gives a factor of $\sqrt{\sigma_n/\sigma_r} \sim 0.01$ for the the parameters reported in our manuscript and in [Nilson, et. al. *Phys. Plasmas* **15**, (2008)], where $\sigma_n < 1$ and $\sigma_r > 1$ are the σ -factor for non-relativistic and relativistic reconnection respectively). Also, in the x -direction (along the reconnection current), the reconnection layer in laser-plasma experiments is typically $100 \mu\text{m} \sim 125d_e$ or $5\rho_0$ (see Fig. 4(a) in [Nilson, et. al. *Phys. Rev. Letts.* **97**, (2006)]), which is smaller than their transverse (inflow and outflow) size of reconnection layer, and clearly not isotropic.

(iii) Finally, while we believe the small system scale reported in our

manuscript is a valid regime to study reconnection, we would like to point out that the system size is determined by the laser spot size (while in x-direction by the scale length of density gradient). A small scale is not essential for the proposed scheme to work. One can obtain a larger system size by paying the price of laser energy. For instance, increasing the laser spot by a factor of 5 gives $L \sim 40d_e$ ($130\rho_0$), and the laser energy would be 5 J (300TW) in this case.

We have added comments covering these points to the manuscript.

3. I need to pick up a point raised by Referee 2, which I do not think the present version adequately addresses - what fundamental physics of relativistic magnetic reconnection can be addressed by this scheme?

Much is known about non-relativistic reconnection, including its role to energize particles. What is novel about magnetic reconnection in relativistic plasmas, which can be addressed by this proposed experiment platform? If a relativistic reconnection simply involves a rescaling of parameters into relativistic counterparts, then what is really new? The authors list several theory publications in which relativistic reconnection is studied - but what controversies or novel aspects of the relativistic regime demand experimental verification? Some of the signatures observed (particle acceleration, agyrotropy and Hall fields) are not unique to relativistic MR.

Reply: We agree that the above mentioned points need to be emphasized. The introduction part of our manuscript has now been revised to address the novelty/importance of the presented results.

(i) The main purpose of this manuscript is to propose a novel laser-plasma-interaction (LPI) based MR scheme. In order to show that reconnection occurs in the presented setup, we analyzed the consistency of the observed signatures with previous MR studies. However, many of these signatures are quantitatively very different to previous LPI based MR studies. In previous LPI based MR studies, judged by typical plasma $\beta \sim 100$, the *total* magnetic energy is $\sim 1\%$ of electron thermal energy that heated by laser, and the released magnetic energy due to MR should be much smaller than this. Such a small amount of energy is very difficult to trace and therefore it is not clear what role MR plays in terms of energy balance. In our proposed setup, the dissipated magnetic energy (~ 3 mJ in the main simulation presented in our manuscript) accounts for 20% of total energy transition (Fig. 4(b) of our manuscript), or 1.5% of the incident laser energy. As the dissipated magnetic

energy constitutes a much larger fraction of the total energy, signatures of reconnection should be easier to detect.

(ii) One of the most interesting topics in relativistic reconnection is the energetics of particle acceleration (considered in the theory publications we listed in our previous response). Some of the predictions can only be tested when the MR associated energy is significant, which is the case in our setup and therefore we believe it is an important advance compared to the current state-of-the-art. For instance, the “kinetic beaming” effect [Cerutti, et. al. *The Astrophys. J.* **754**, (2012)], where the most energetic particles form compact bunches in the reconnection layer, can be difficult to test in a high- β experimental setup where the most energetic particles might be produced by the other sources (e.g. laser heating of plasma) and have nothing to do with MR. For example in the case presented in [Raymond, et. al. *arXiv:1610.06866* (2016)], it can be seen in their Fig. 2(d) that the MR-associated particles create a bump on the thermal background, but the cut-off energies are quite similar. The “kinetic beaming” effect is one of the novel aspects of relativistic reconnection that could be tested with the presented scheme, since it’s a result of relativistic particles under acceleration in the reconnection electric field and deflected by the reconnected magnetic fields [Uzdensky, et. al. *The Astrophys. J. Letts.* **737** (2011)], which does not rely crucially on the onset of reconnection.

Another fundamental problem in relativistic reconnection is that so far most of the simulation studies have been performed for pair-plasmas or artificial electron-ion mass. Since in real experiments the electron-ion plasma is used, our setup could also help to study the role that ions play in this process.

(iii) Relativistic reconnection is not just simply a rescaling of parameters. The resulting power-law electron spectrum with hard slopes that are close to unity for large σ is one of the most interesting features that attracts extensive attention. Many studies have been done and many distinct acceleration mechanisms have been identified, however the majority of the studies use idealized initial conditions such as the kinetic Harris equilibrium as well as a preformed current sheet. Recent simulations without preformed current sheets result in a significantly softer spectral index [Nalewajko, et. al. *The Astrophys. J.* **826**, (2016)]. Therefore the study of the formation of such power-law spectra remains an open area of research, in particular how it depends on the plasma conditions. Our scheme will provide a possibility to study this process in the laboratory, albeit caution should be taken as the relative importance of different acceleration mechanisms rely on the system sizes [Dahlin, et. al. *Phys. Plasmas* **21**, (2014)].

(iv) Examples of some of the interesting physics associated with *strongly-driven* MR that can potentially be studied with the proposed experimental platform are the following: (a) particle acceleration in the driven relativistic MR [see Lyubarsky, et. al. *The Astrophys. J.* **682**, (2008)], the spectra of particles initially at different distances from the X-point could be studied and compared with the reported simulation results. (Again, the low- β regime is important for this kind of study, otherwise the difference in the electron spectra is drowned by a thermal distribution; and a laser with large spot size should be used.) (b) reconnection rate in strongly-driven MR [see Birn, et. al. *Phys. Plasma* **14**, (2007)], the reported linear dependence on external driving force (maybe modified by the separation of laser-driven electron beams) can be tested in the relativistic MR regime.

The studies listed in (ii) \sim (iv) are expected to greatly advance the understanding of relativistic MR. However, they are extensive studies in themselves and beyond the scope of the current manuscript.

Thank you again for your comments! We hope that with our clarifications and amendments you can recommend the paper for publication.

Sincerely Yours,
The Authors

REVIEWERS' COMMENTS:

Reviewer #2 (Remarks to the Author):

The authors have responded to my questions in a satisfactory manner and made some changes in the manuscript that improved it. I would just note that while it is clear that current laser-driven experiments in the non-relativistic regime operate in a high β regime, it is not clear the same will be true at relativistic intensities. The estimate $n_h T_h$ given by the authors in their response is the fast electron energy density produced by the laser in the interaction and not the thermal electron energy density as a result of expansion. The paper by Macchi et al is very clear in this regard. This means that $\beta = 1-3$ is at the laser-plasma region. A fraction of fast electrons will propagate through the target and leave the target. For reconnection, what matters are the hot electrons which propagate/expand radially along the near-critical density surface, which are a fraction of the number produced and their density will drop quickly with radius. I am thus still not fully convinced that more standard configurations would not allow the study of this regime with a simpler setup. In any case, that will certainly be the focus of future works and this paper will stimulate such studies both on the simulation and experimental side. Therefore, I recommend publication.

Reviewer #3 (Remarks to the Author):

The authors have appropriately answered my questions, and have amended the discussion in the paper of the physics of the system size the value of studying relativistic reconnection in the laboratory.

If all referees are satisfied I will concur to recommend publication.

Response to Reviewer 2

The authors have responded to my questions in a satisfactory manner and made some changes in the manuscript that improved it. I would just note that while it is clear that current laser-driven experiments in the non-relativistic regime operate in a high β regime, it is not clear the same will be true at relativistic intensities. The estimate $n_h T_h$ given by the authors in their response is the fast electron energy density produced by the laser in the interaction and not the thermal electron energy density as a result of expansion. The paper by Macchi et al is very clear in this regard. This means that $\beta = 1-3$ is at the laser-plasma region. A fraction of fast electrons will propagate through the target and leave the target. For reconnection, what matters are the hot electrons which propagate/expand radially along the near-critical density surface, which are a fraction of the number produced and their density will drop quickly with radius. I am thus still not fully convinced that more standard configurations would not allow the study of this regime with a simpler setup. In any case, that will certainly be the focus of future works and this paper will stimulate such studies both on the simulation and experimental side. Therefore, I recommend publication.

Reply: We sincerely appreciate your careful reading of our manuscript and your recommendation for publication. Regarding the estimation of β , which is the ratio of the electron kinetic energy density and the magnetic energy density, we believe that since the magnetic field expands with the hot electrons, it is reasonable to assume the volume of hot electrons and magnetic field is of similar size. Therefore, what matters is the fraction of laser energy that is transferred to electron energy and magnetic energy.

As we argued in the manuscript, according to previous experiments, the laser-heating of electrons is very efficient at relativistic intensities. An empirical scaling law $\eta \approx 1.2 \times 10^{-15} I^{0.74}$ [Fuchs, et al. *Nat. Phys.* **2**, (2006)] indicates that the ratio of laser energy converted to the hot electrons (η) approaches 50% as the laser intensity reaches $5 \times 10^{19} \text{ W cm}^{-2}$, which means that achieving $\beta < 1$ would be impossible. Moreover, even if $\eta < 50\%$ for laser intensities below $5 \times 10^{19} \text{ W cm}^{-2}$, since there is still large amount of laser energy that is reflected, transfer to the electrostatic field (which is usually higher or at least the same as the magnetic field), ion kinetic energy, and so on, it would still be very challenging to achieve $\beta < 1$.

Thank you again for your constructive comments.

Sincerely Yours,
The Authors